# Frequency-dependent selection of neoantigens fosters tumor immune escape and predicts immunotherapy response

Shaoqing Chen[1,2,3,10], Duo Xie [2,4,10], Zan Li[5], Jiguang Wang[6,7,8,9], Zheng Hu [2] ✉ & Da Zhou [1,3] ✉

Cancer is an evolutionary process shaped by selective pressure from the microenvironments. However, recent studies reveal that certain tumors undergo neutral evolution where there is no detectable fitness difference amongst the cells following malignant transformation. Here, through computational modeling, we demonstrate that negative frequency-dependent selection (or NFDS), where the immune response against cancer cells depends on the clonality of neoantigens, can lead to an immunogenic landscape that is highly similar to neutral evolution. Crucially, NFDS promotes high antigenic heterogeneity and early immune evasion in hypermutable tumors, leading to poor responses to immune checkpoint blockade (ICB) therapy. Our model also reveals that NFDS is characterized by a negative association between average clonality and total burden of neoantigens. Indeed, this unique feature of NFDS is common in the whole-exome sequencing (WES) datasets (357 tumor samples from 275 patients) from four melanoma cohorts with ICB therapy and a non-small cell lung cancer (NSCLC) WES dataset (327 tumor samples from 100 patients). Altogether, our study provides quantitative evidence supporting the theory of NFDS in cancer, explaining the high prevalence of neutral-looking tumors. These findings also highlight the critical role of frequency-dependent selection in devising more efficient and predictive immunotherapies.

Cancer is an evolutionary process driven by genetic or epigenetic alterations in the genome and the interactions between cancer cells and the microenvironments[1–4]. Understanding the evolutionary dynamics that are operative at different stages of progression in individual tumors might inform the earlier detection, diagnosis, and treatment of cancer[5]. Cancer evolution is particularly shaped by a complex interplay between mutated cancer cells and immune cells capable of recognizing and combating them[6]. Herein, somatic mutations can be translated into novel peptides and presented on the cancer cell surface, thus generating neoantigens[7]. Tumor neoantigens are capable of eliciting a T-cell response and are presented by

the patient's human leukocyte antigen (HLA) molecules, prompting the T cells to attack and eliminate the mutated cancer cells[8].

This antitumor immune response confers a negative selection pressure[9], and there are several mechanisms that cancers can co-opt to evade immune recognition and/or elimination[10,11]. For instance, the programmed death ligand 1 (PD-L1) can inhibit the immune system by interacting with PD-1 on activated T cells, B cells, and myeloid cells[12]. This understanding has fueled the development of immune checkpoint blockade (ICB) therapies, which have shown significant promise in enhancing the survival rates of patients with a variety of tumor types[13]. However, certain tumor patients

[1]School of Mathematical Sciences, Xiamen University, Xiamen, China. [2]Key Laboratory of Quantitative Synthetic Biology, Shenzhen Institute of Synthetic Biology, Shenzhen Institute of Advanced Technology, Chinese Academy of Sciences, Shenzhen, China. [3]National Institute for Data Science in Health and Medicine, Xiamen University, Xiamen, China. [4]Faculty of Health Sciences, University of Macau, Taipa, Macau, China. [5]Life Science Research Center, Core Research Facilities, Southern University of Science and Technology, Shenzhen, China. [6]Division of Life Science and State Key Laboratory of Molecular Neuroscience, The Hong Kong University of Science and Technology, Clear Water Bay, Kowloon, Hong Kong SAR, China. [7]Department of Chemical and Biological Engineering, The Hong Kong University of Science and Technology, Clear Water Bay, Kowloon, Hong Kong SAR, China. [8]Hong Kong Center for Neurodegenerative Diseases, InnoHK, Hong Kong SAR, China. [9]HKUST Shenzhen-Hong Kong Collaborative Innovation Research Institute, Futian, Shenzhen, China. [10]These authors contributed equally: Shaoqing Chen, Duo Xie. ✉e-mail: zheng.hu@siat.ac.cn; zhouda@xmu.edu.cn

remain unresponsive to ICB therapies despite high tumor mutational burden and/or extensive immune cell infiltration, while the underlying mechanisms remain elusive[14,15]. Several explanations have been proposed, such as insufficient immune cell infiltration[16], PD-L1 overexpression offsetting therapy[17], loss of an HLA haplotype[18,19], somatic mutations in B2M gene or HLA alleles leading to dysfunctional neoantigen presentation[20,21], high intra-tumor heterogeneity[22], *etc.*

Several studies have indicated that certain types of tumors can evolve neutrally, wherein subclonal lineages exhibit no selective difference in cell fitness[23–26]. Interestingly, these neutral tumors are often linked to immune escape and poor prognosis[27]. In fact, neutral tumor evolution leads to high intra-tumor heterogeneity (ITH)[23] and thus may result in an ineffective immune response of the developing tumor[28]. However, clonal selection has also been found in patients with poor prognosis[29], challenging the view of neutral evolution[23–26]. The limitations of these evolutionary features necessitate in-depth investigation of immune selection in tumors.

Noting the parallels between cancer-immune dynamics and prey-predator interactions in ecological systems[30,31], here we propose that immune-driven cancer evolution adheres to organism evolution under negative frequency-dependent selection (NFDS), leading to effectively neutral evolution. Essentially, predators are inclined to prey on the most abundant species, inadvertently providing survival advantages to less abundant species[30]. Analogously, immune cells could target cancer cells expressing a certain abundance of neoantigens allowing those with fewer neoantigens to dodge immune responses. In fact, mouse experiments have shown that immunogenic tumor antigens do not lead to immune-mediated cell rejection when the fraction of cells bearing each antigen ('clonal fraction') is low[32], thus supporting NFDS in immune-driven cancer evolution. However, this theory remains to be tested in human patient samples.

In this study, we employed stochastic modeling to investigate the impact of NFDS on the evolutionary dynamics and neoantigen subclonal structure of hypermutated tumors. We found that tumors subject to NFDS exhibit high tumor heterogeneity and low clonality of neoantigens. We then simulated tumor immunotherapy and discovered that NFDS tumors were less responsive to ICB therapies than tumors under conventional negative selection (NS). Furthermore, pre-therapy NS and NFDS *virtual* tumors exhibited distinct features regarding the clonality of neoantigens, which associated with different ICB therapy responses. We then analyzed WES datasets from four melanoma patient cohorts including 357 tumor samples from 275 patients[16,22,33,34] and one non-small cell lung cancer (NSCLC) cohort including 327 tumor samples from 100 patients[35], and found that NFDS was supported by the unique evolutionary patterns in these datasets.

## Results

### Modeling neoantigen evolution under different immune selection scenarios

We utilized a stochastic branching model to investigate tumor evolution under four immune selection scenarios: (1) negative selection (NS); (2) NFDS; (3) NS with subclonal immune escape (IE); and (4) NFDS with subclonal immune escape (IE). A cancer cell harboring a sufficient amount of neoantigens is defined as immunogenic. While both NS and NFDS undergo negative Darwinian selection (selective coefficient $s<0$) against immunogenic cancer cells, NS allows the immune system to respond even if only one immunogenic cell exists, independent of the frequency of immunogenic cells in a population[9,36,37]. In contrast, NFDS requires a sufficient number of immunogenic cells to trigger an unremitting immune response (Fig. 1a). Here highly immunogenic neoantigens that present in high clonality (measured by cancer cell fraction or CCF) have a higher probability of meeting both cutoffs we set, and thus will be more susceptible to negative selection[38] (see **Methods**). For the remaining two scenarios of immune selection, we considered subclonal immune escape upon the NS or NFDS by allowing cancer cells to stochastically acquire immune escaped alterations, leading to neutral evolution of the immune escaped lineages (selective coefficient $s = 0$), regardless of the cell's antigenicity (Fig. 1b, **Methods**).

The stochastic branching model of the tumor growth is initiated with an antigenically neutral cancer cell (Fig. 1b). A cancer cell $i$ can divide (with a division probability $b$), die (with a death rate $d_{C_i}$), or enter a quiescent state (quiescent probability is $1 - b - d_{C_i}$; cells remain viable but stop dividing). Each cell division resulted in $m$ unique mutations that follow a Poisson distribution, $m \sim$ Poisson $(p_m)$. The newly occurred mutations were neoantigens at rate $p$, or as passengers (evolutionarily neutral) at rate $1 - p$. Each neoantigen exhibited an antigenicity score $(a_k^i)$ drawn from an exponential distribution. The tumor cell death rate $d_{C_i}$ was determined by the cumulative antigenicity, i.e. $A_{C_i} = \sum a_k^i$ and the negative selection intensity $s$ as Eq. (1) (**Methods**). Here $A_{C_i}$ was defined as the sum of the antigenicity of neoantigens harbored in the lineage. NFDS was modeled by setting a threshold of tumor immunogenicity, $\frac{T_{A_C}}{T} > c_2$, where $T_{A_C}$ and $T$ were the number of immunogenic tumor cells and total number of tumor cells, respectively (Fig. 1b, **Methods**).

With this model, we first examined the growth dynamics of NS and NFDS tumors, respectively (Fig. 1c, d). We found that NFDS tumors grew faster than NS tumors, indicating that NS imposed a stronger suppression on tumor growth than NFDS. As selection intensified, NS tumors grew more slowly, while NFDS tumors still maintained a steady growth rate (Fig. 1d), indicating that NFDS tended to protect the cancer cells from elimination even under stringent negative selection ($s = -0.8$). We demonstrated that the difference in growth rates between NS and NFDS is amplified by higher neoantigen acquisition rate and stronger negative selection (Supplementary Fig. 1). Furthermore, NFDS tumors grow faster than NS tumors with subclonal immune escape, driven by factors such as neoantigen acquisition and negative selection (Supplementary Fig. 2). However, they display indistinguishable growth dynamics when experiencing clonal immune escape (Supplementary Fig. 3).

We then examined the antigenic mutation landscape under each of four scenarios of immune selection. As expected, the simulation results showed that the number of antigenic mutations in a NS tumor decreased as selection intensified (Fig. 1e). Interestingly, although NFDS tumors had a lower antigenic mutation burden as selection intensity increased from low to moderate (Fig. 1e), further increases in selection intensity ($s = -1.2$) did not significantly reduce neoantigen burden, but rather increased it. These patterns were observed across different cancer cell fraction (CCF) thresholds as well as different cell and tumor immunogenicity thresholds (Supplementary Figs. 4–6). Tumors with subclonal immune escape showed similar rollback patterns as NFDS (Fig. 1f) because mutations accumulate rapidly in the immune-escaped lineages following neutral evolution. However, the cause of the same rollback pattern was different for NFDS where low-cellularity neoantigens of immunogenic cells were maintained by unsustainable negative selection. Our simulations indicated that tumors experienced interleaved immune activation and suppression under intense NFDS (Supplementary Fig. 7b). This mechanism is similar to immune exhaustion, where the immune system faces constant pathogen infections and inflammations[39]. The amplitude and frequency of the oscillation determine the profound influence of NFDS on antigenic mutation accumulation, with a more significant effect observed under a relatively weaker selection intensity. We also observed this effect from the difference in growth rates between NS tumors and NFDS tumors (Supplementary Fig. 7c, d).

This finding is intriguing because the pattern of neoantigens accumulation under NFDS resembles subclonal immune escape. It is widely accepted that tumor immune escape is achieved through specific genomic mutations in the genome[9,40–42]. However, our results suggest that the specific interactions between tumor cells and the immune microenvironment confer immune-escape-like evolutionary dynamics.

### NFDS promotes evolutionary rescue in hypermutated tumors

We explored the effect of subclonal immune escape on tumor growth. Unlike the growth patterns in NFDS, NS tumor showed an initial slowdown in growth, followed by an accelerated pace upon the random acquisition of

immune escape alterations (Fig. 2a). Subclonal immune escape has significant effect on NS tumor growth compared to its diminished role under NFDS (Fig. 2b). To systematically compare the evolutionary dynamics of four immune selection scenarios, we simulated 20 tumors with varying selection intensity and mutation rates (Supplementary Figs. 8–10). Notably,

all NS cancer cells without immune escape were eventually eliminated under high mutation rate ($\mu = 5.5$ per cell division), and all NFDS tumors were able to grow to the maximum population size (100,000 cells) (Supplementary Fig. 8c). This significant contrast remains robust across various levels of selection intensity. (Supplementary Figs. 9b and 10b). We also

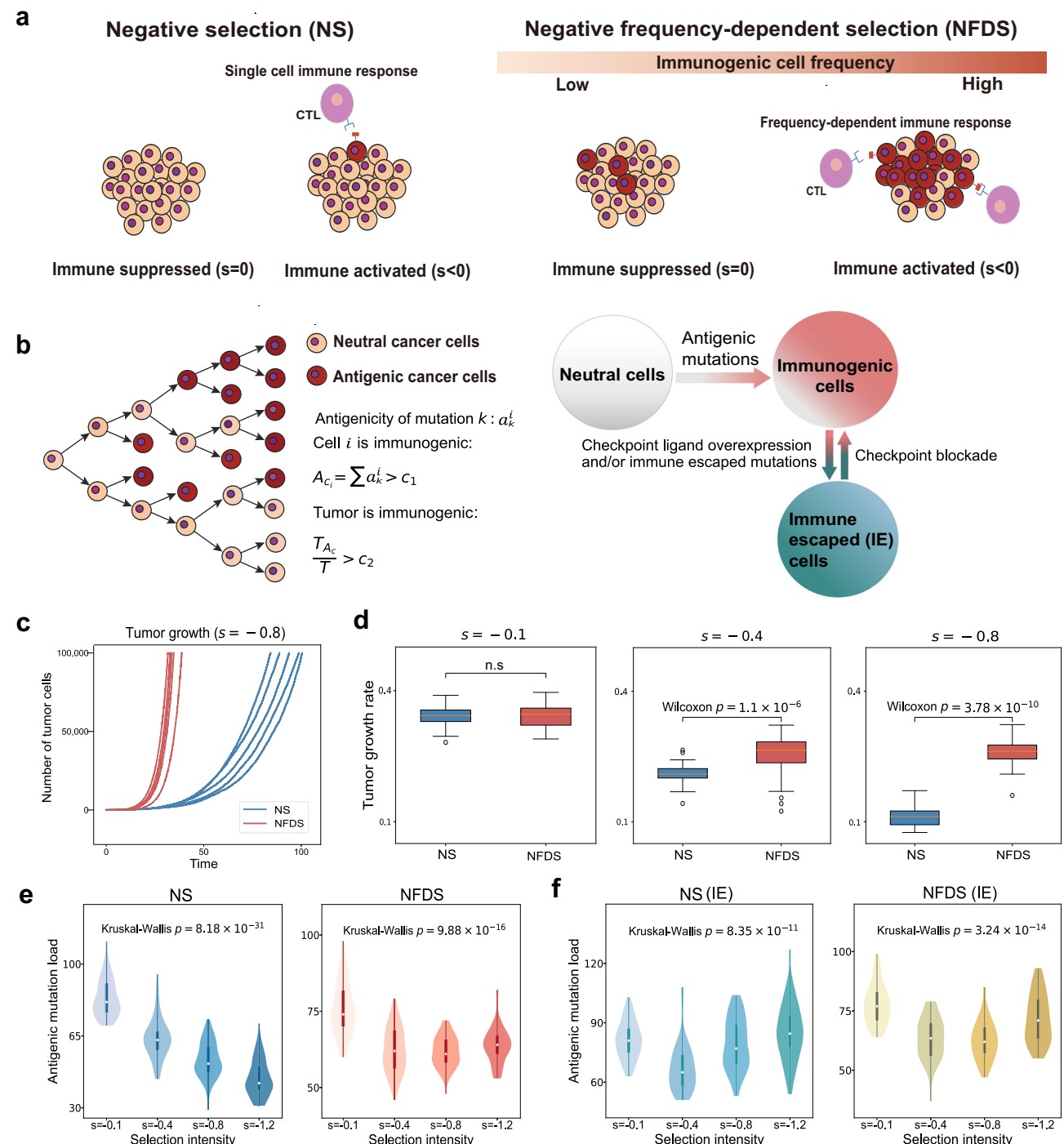

**Fig. 1 | Models of neoantigen-driven tumor evolution. a** Schematic of negative selection (NS) and negative frequency-dependent selection (NFDS). The red-filled circles represent antigenic cancer cells and purple-filled circles represent cytotoxic T lymphocytes (CTLs). **b** Schematic of stochastic branching process with accumulation of neutral and antigenic mutations. $c_1$ is the threshold of cell immunogenicity while $c_2$ is the threshold of tumor immunogenicity. $T_{A_C}$ and $T$ are the number of immunogenic and total number of tumor cells, respectively. **c** Growth curves of five simulated tumors under NS and NFDS, respectively with selection coefficient $s = -0.8$. **d** Tumor growth rates of simulated tumors ($n = 50$) under NS and NFDS,

respectively at varying selective intensities. Each simulation was terminated when the tumors reached 100,000 cells. Box plots show median, quartiles (boxes) and range (whiskers). $p$ values, Wilcoxon rank-sum one-sided test. **e** Number of neoantigens accumulated in simulated tumors at varying selective intensities (each with 50 simulations). **f** Number of neoantigens accumulated in simulated tumors with subclonal immune escape (IE, immune escape probability $p_e = 10^{-4}$) at varying selective intensities (each with 50 simulations). All tumors were simulated with a tumor mutation rate of $\mu = 5$ per cell division per genome. Violin plots show median, quartiles, and range (whiskers). $p$ values, the Kruskal-Wallis test.

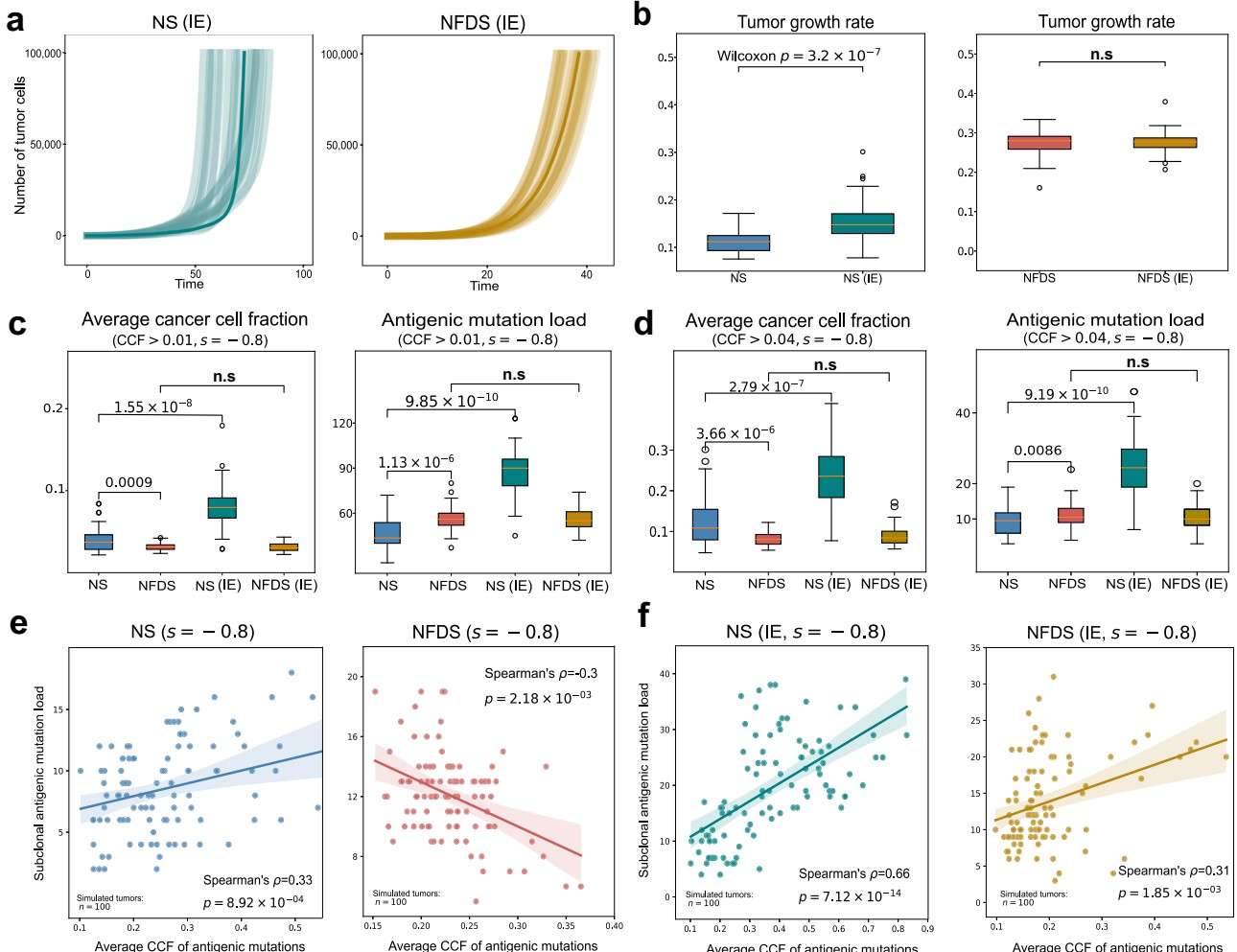

**Fig. 2 | Effect of IE on tumor growth and characteristic neoantigen landscape of NFDS. a** Growth curves of 20 simulated tumors under NS (IE) and NFDS (IE), respectively. **b** Tumor growth rates of simulated tumors ($n = 50$) under NS and NFDS, respectively with and without IE. Box plots showing the antigenic mutations accumulated in 50 simulated tumors with cancer cell fraction (CCF) > 0.01 (**c**) and CCF > 0.04 (**d**), respectively. The average CCF and mutation load of antigenic mutations from 50 simulated tumors under four evolutionary scenarios are shown. Box plots show median, quartiles (boxes) and range (whiskers). **e** Correlation analysis between average CCF and subclonal antigenic mutation load of 100 simulated tumors undergoing NS and NFDS (CCF > 0.1), respectively. **f** Correlation analysis between average CCF and subclonal antigenic mutation load of 100 simulated tumors undergoing NS and NFDS with subclonal immune (immune escape probability $p_e = 10^{-4}$, CCF > 0.1), respectively. The line indicates the linear regression and the shading indicates the 95% confidence interval (CI) of the regression. All tumors were simulated with a tumor mutation rate of $\mu = 5$ per cell division per genome. $p$ values, Wilcoxon rank-sum one-sided test.

observed that *virtual* tumors with subclonal immune escape experienced evolutionary rescue and all grew to the maximum population size (Supplementary Fig. 8d). These results suggest that NFDS is effectively lower selection against immunogenicity.

However, when the mutation rate was a little higher ($\mu = 6$ per cell division), all *virtual* tumors without subclonal immune escape were eventually eliminated by intense negative selection ($s \geq -0.4$), although most NFDS tumors reached a larger population size than NS tumors (Supplementary Figs. 8e, 9c and 10c). In contrast, tumors with subclonal immune escape still grew to the maximum population size (Supplementary Fig. 8f). In addition, we considered driver events that can be immunogenic (**Methods**) and discovered evolutionary rescue in NS tumors with subclonal driver mutations at high mutation rate ($\mu = 5.5$) (Supplementary Figs. 9d). Under stronger negative selection and higher mutation rate, tumors are still eliminated (Supplementary Figs. 9e). NFDS tumors are more susceptible to driver events at high mutation rate and intense negative selection (Supplementary Figs. 10d). The extinction of tumors recapitulated Muller's ratchet-like effect of mutation-driven meltdown of a population[43].

To present a comprehensive perspective, we explore the effect of IE on growth with various combinations of $\mu$ and $s$ (Supplementary Fig. 11). The pronounced influence of IE on NS tumors contrasts with its nonsignificant effect under NFDS conditions (Supplementary Fig. 11a and b). Subclonal immune escape increases tumor growth rate with the effect being exacerbated under elevated mutation rates and negative selection intensity (Supplementary Fig. 11). These results suggest, on the one hand, that the evolutionary rescue conferred by NFDS is dependent on the mutation rate. On the other hand, NFDS may coexist with conventional immune escape mechanisms (e.g., escape mutations) in tumors with a high mutation rate.

Given that NFDS can diversify the ecological system in species evolution[44–46], we next ask whether this type of balancing selection will increase the clonal diversity of antigenic mutations in cancer cells. To address this question, we characterized the antigenic mutation load, clonality, and Shannon diversity of virtual tumors in different scenarios. As expected, NFDS tumors harbored higher antigenic mutation burden and clonal diversity as compared to the conventional negative selection (Fig. 2c; Supplementary Fig. 12a), demonstrating that NFDS increases the diversity of subclonal neoantigens. We also noted that the average CCFs of antigenic mutations in NFDS tumors were significantly lower than in NS tumors (Fig. 2c). This is because NFDS tended to remove high-frequency neoantigens[38]. Subclonal immune escape increased the average

neoantigen CCF, antigenic mutation load and Shannon diversity of NS tumors but not for NFDS tumors (Fig. 2c-d, Supplementary Fig. 12a, b), suggesting that further acquirement of immune escape alterations has little impact on the neoantigen landscape in NFDS tumors. We also found that tumors under NS eventually had larger fractions of immune escaped cells than ones under NFDS (Supplementary Fig. 12c). As higher fraction of escaped cells associates with larger population size, NFDS tumors have to be larger than NS tumors to acquire comparable prevalence of escaped cells (Supplementary Fig. 13). These results suggest that subclonal immune escape has a significant influence on the antigenic mutation landscape under NS but little influence under NFDS (Fig. 2c, d; Supplementary Figs. 12–14).

To test whether NS and NFDS tumors have different evolutionary patterns of neoantigens, we then simulated 100 *virtual* tumors for each of four scenarios and calculated the antigenic mutation load and average CCF for each *virtual* tumor. We found that in NS tumors, the average neoantigen CCF was positively associated with antigenic mutation load (Fig. 2e). In contrast, this association is negative in NFDS tumors, regardless of cell-immunogenic or tumor-immunogenic thresholds (Fig. 2e; Supplementary Fig. 15). However, in all tumors with subclonal immune escape, we also observed a positive correlation (Fig. 2f). The identified correlations depend on the selection intensity $s$, with the correlations becoming more significant as the selection intensifies. (Supplementary Fig. 16, 17). To investigate the underlying mechanisms of these correlations, we then performed a mathematical analysis using the ordinary differential equation (ODE) model. According to Eqs. (4) and (7) in Supplementary Note, the correlations between the average CCF and the number of antigenic mutations can be inferred, which confirmed the negative correlations between the average neoantigen CCF and the antigenic mutation load under NFDS (Figs. 2e and 2f).

In NS tumors with immune escape, we also found that the average CCF was strongly correlated with the proportion of immune escaped cells (Supplementary Fig. 18a). However, this correlation was weaker in NFDS tumors with immune escape (Supplementary Fig. 18b), suggesting that in NFDS tumors, higher intra-tumoral antigenic heterogeneity provides the potential for greater antigenic mutation burden at lower frequencies. In contrast, in tumors subject to intense negative selection, a higher antigenic mutation load typically arises from earlier immune escape, resulting in a higher average CCF of antigenic mutations. These observations suggest that tumors may adopt different strategies to maintain a substantial burden of antigenic mutations. NFDS is effectively lower selection for genetic immune escape. Importantly, the distinct association between average neoantigen CCF and antigenic mutation load offers a way to test NFDS in real patient data.

## Lack of response to ICB therapy in NFDS *virtual* tumors

To further explore the impact of immune checkpoint blockade (ICB) therapy on NS and NFDS tumors, respectively, we modeled ICB therapy by restoring negative selection on immune escaped cells. We generated 50 treated and 50 untreated virtual patients with NS and NFDS tumors, respectively. We defined a *virtual* patient as deceased when the tumor reached a maximum cell population size (100,000 cells). Survival analysis revealed that NS patients experience significantly longer overall survival after treatment (Fig. 3a), whereas NFDS patients did not show a significant benefit from ICB (Fig. 3b). Simulations showed that NS tumors can be eliminated by the immune system after ICB treatment, which can be classified as "responders" (Supplementary Fig. 19a). In contrast, although NFDS tumors shrank earlier in size after ICB therapy, they eventually manifested as progressive disease and therefore can be classified as "non-responders" (Supplementary Fig. 19a). Survival analysis showed that ICB-treated *virtual* patients with NFDS tumors had significantly lower survival rates than patients with NS tumors (Fig. 3c). In untreated virtual patients, NFDS tumors still exhibited worse survival (Fig. 3d), suggesting that NFDS tumors may represent a more aggressive subtype. These results were also consistent with the clinical observations that heterogeneous tumors have a worse response to immunotherapy[15,28,34]. Notably, the negative correlation between average CCF and subclonal antigenic mutation load was maintained even after ICB therapy (Supplementary Fig. 19b). Furthermore, we

can observe the benefit from ICB, which can be evaluated with survival analysis of virtual patients, is related to the effect of immune escape (Supplementary Figs. 11 and 20).

We observed characteristic CCF distributions with distinct subclones in untreated NS tumors (Fig. 3e and supplementary Fig. 21). However, these distinct subclones were not found in untreated NFDS tumors, in line with effectively neutral evolution[47] (Fig. 3e and supplementary Fig. 21). By fitting the CCF distributions of simulated pre-therapy tumors (virtual tumors with subclonal immune escape) to the power-law model of neutral evolution[25], we found a significantly higher $R^2$ fitting score in NFDS tumors as compared to NS tumors (Fig. 3f). The difference between $R^2$ values for NS and NFDS depends on the influence of immune escape. A significant difference in $R^2$ values applies in tumors with early subclonal immune escape and strong selection intensities (Supplementary Fig. 22). The results were similar when taking into account varying sample sequencing depths and tumor purities (Supplementary Fig. 23). Of note, for NS tumors, the average CCF was significantly decreased after immunotherapy (Fig. 3g), while the average CCF of NFDS tumors was only slightly affected by immunotherapy (Fig. 3h). Taken together, these results suggest that the pre-therapy CCF distributions might inform the prognosis of immunotherapy. Namely, tumors with a CCF distribution containing distinct high-frequency subclones may respond better to immunotherapy.

## Predicting response to ICB therapy based on pre-therapy CCF distributions

Next, we sought to test NFDS in real-world patients. In particular, our simulations have generated a testable prediction for NFDS, namely the negative correlation between the average CCF of neoantigens and subclonal antigenic mutation load. We analyzed whole-exome sequencing (WES) data from tumor biopsies of four publicly available melanoma patients with ICB therapy[16,22,33,34] and one non-small cell lung cancer (NSCLC) cohort of 100 patients[35]. Since melanoma generally exhibits a high mutation burden[48], these sequenced biopsies provide an excellent dataset to test our model. An in-house bioinformatics pipeline[49,50] was used to identify somatic single nucleotide variants (sSNVs), insertions/deletions (indels) and somatic copy number alterations (sCNAs). We then predicted neoantigens based on amino acid changes due to sSNVs and indels using NeoPredPipe[51] (**Methods**). We also estimated the cancer cell fraction (CCF) of sSNVs and indels to discriminate clonal (the upper bound of the 95% confidence interval (CI) of CCF ≥ 1) versus subclonal (the upper bound of the 95% CI of CCF < 1) sSNVs and indels (**Methods**).

Indeed, we found a significant negative correlation between the average CCF of neoantigens and the number of subclonal antigenic mutations in melanoma patients from Reuben et al[34]. (Fig. 4a), indicating this dataset fits well with the expectation of NFDS model[33]. We also observed a similar negative correlation in the pre-therapy patient cohort of Amato et al. (Fig. 4b). To investigate the effect of immune escape, we analyzed post-therapy non-responder samples (labeled as stable disease/progression disease (SD/PD) according to RECIST 1.1 criteria[52]) from this cohort and also found a trend of negative correlation between the two variables (Spearman's $\rho = -0.90$, Fig. 4c). In another cohort Riaz et al[16]., we found a significant positive correlation in both pre- and post-therapy samples (Fig. 4d-e), which rejected NFDS model. However, when focusing on the post-therapy non-responder samples (SD/PD) with high mutation rate (subclonal neoantigen burden $N > 50$), a negative trend (Spearman's $\rho = -0.62$) between average CCF and subclonal neoantigen burden was still observed, although not statistically significant due to limited sample size (Fig. 4f). These results support NFDS of neoantigens in hypermutated melanoma tumors after ICB immunotherapy. We also examined the NFDS patterns (Supplementary Fig. 24) using all subclonal mutations without neoantigen prediction. We found a consistent pattern with the one obtained using neoantigen mutations (Fig. 4a–f), indicating that the negative correlation pattern of subclonal neoantigen mutations can be inferred from all subclonal mutations.

One of the well-established predictors of response to ICB therapy is T-cell infiltration, as also shown in the Riaz cohort[16] (Supplementary

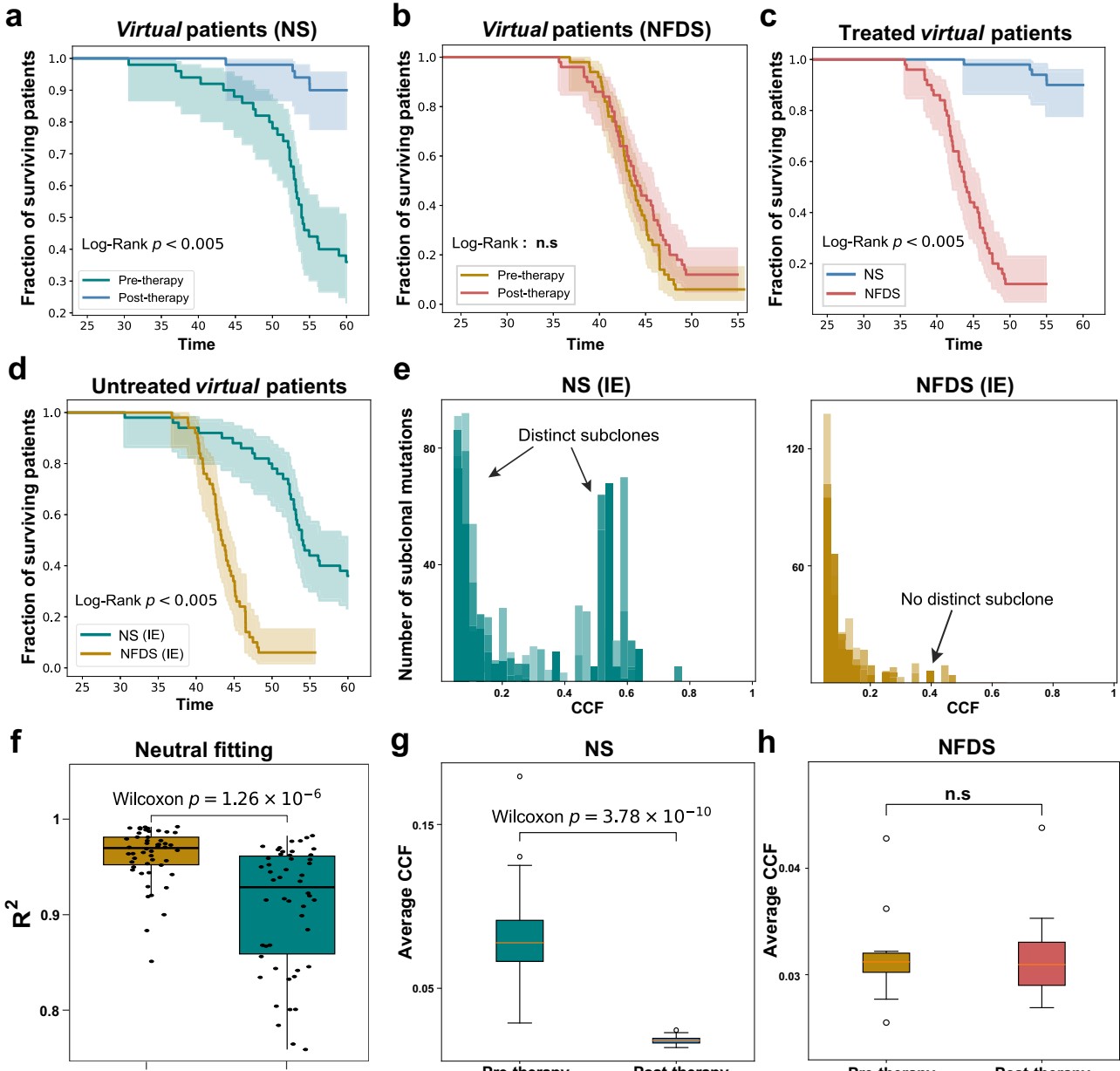

**Fig. 3 | Computational simulations of immune checkpoint blockade (ICB) therapy.** **a, b** Kaplan-Meier curves of 50 *virtual* patients before and after immunotherapy. *Virtual* patients receive ICB therapy at simulation time of 25. **c** Kaplan-Meier curves of 50 treated *virtual* patients. **d** Kaplan-Meier curves of 50 untreated *virtual* patients. **e** CCF distributions of all mutations (including neutral passengers) of five simulated tumors under immune escape mode of NS and NFDS, respectively. **f** Identification of neutrality for simulated pre-therapy tumors. (with subclonal

immune escape or IE). The $R^2$ represents degree of fitting to power-law distribution. Average CCF of simulated tumors that underwent NS (**g**) and NFDS (**h**), respectively before and after ICB therapy. Box plots show median, quartiles (boxes), and range (whiskers). The model parameters used are immune escape probability $p_e = 10^{-4}$, selection coefficient $s = -0.8$ and mutation rate $\mu = 5$. $p$ values, Wilcoxon rank-sum one-sided test.

Fig. 25a). However, our results suggest the existence of another effective biomarker when T-cell infiltration is comparable between responders (partial response /complete response, PR/CR) and non-responders (SD/PD), that is the evolutionary mode of pre-therapy tumors (Fig. 3c, d, e, f). In particular, we observed no significant difference in cytolytic scores (representing T-cell infiltration) between responders and non-responders in Liu et al.'s cohort[22] (Supplementary Fig. 25b). However, through neutrality testing on this dataset, a significant difference in $R^2$ values between responders and non-responders was observed (Fig. 4g and Supplementary Fig. 26). This result was consistent with our model prediction that neutral-like evolution is associated with worse response to ICB therapy (Fig. 3c, d, e, f). This pattern is mirrored when performing the same test in Amato et al.'s cohort[33] (Fig. 4h

and Supplementary Fig. 27). As expected, we identified a significant positive correlation in the Riaz cohort[16] (Fig. 4d, e). However, this correlation did not extend to the NFDS model, as no significant difference was found in the $R^2$ values between responders and non-responders (Supplementary Fig. 25c). To evaluate the clinical significance of neutrality in CCF distributions, we set a threshold for high $R^2$ values and performed survival analysis on patients from Liu et al.'s cohort where the patient survival information was available[22]. Indeed, patients with high $R^2$ values (more neutral-like) showed a significantly shorter overall survival as compared to patients with low $R^2$ values (more selection-like) (Fig. 4i). We then used an external validation cohort[35] comprising 100 treated lung tumor patients to explore the role of NFDS in immune evasion across different tumor types. In the validation

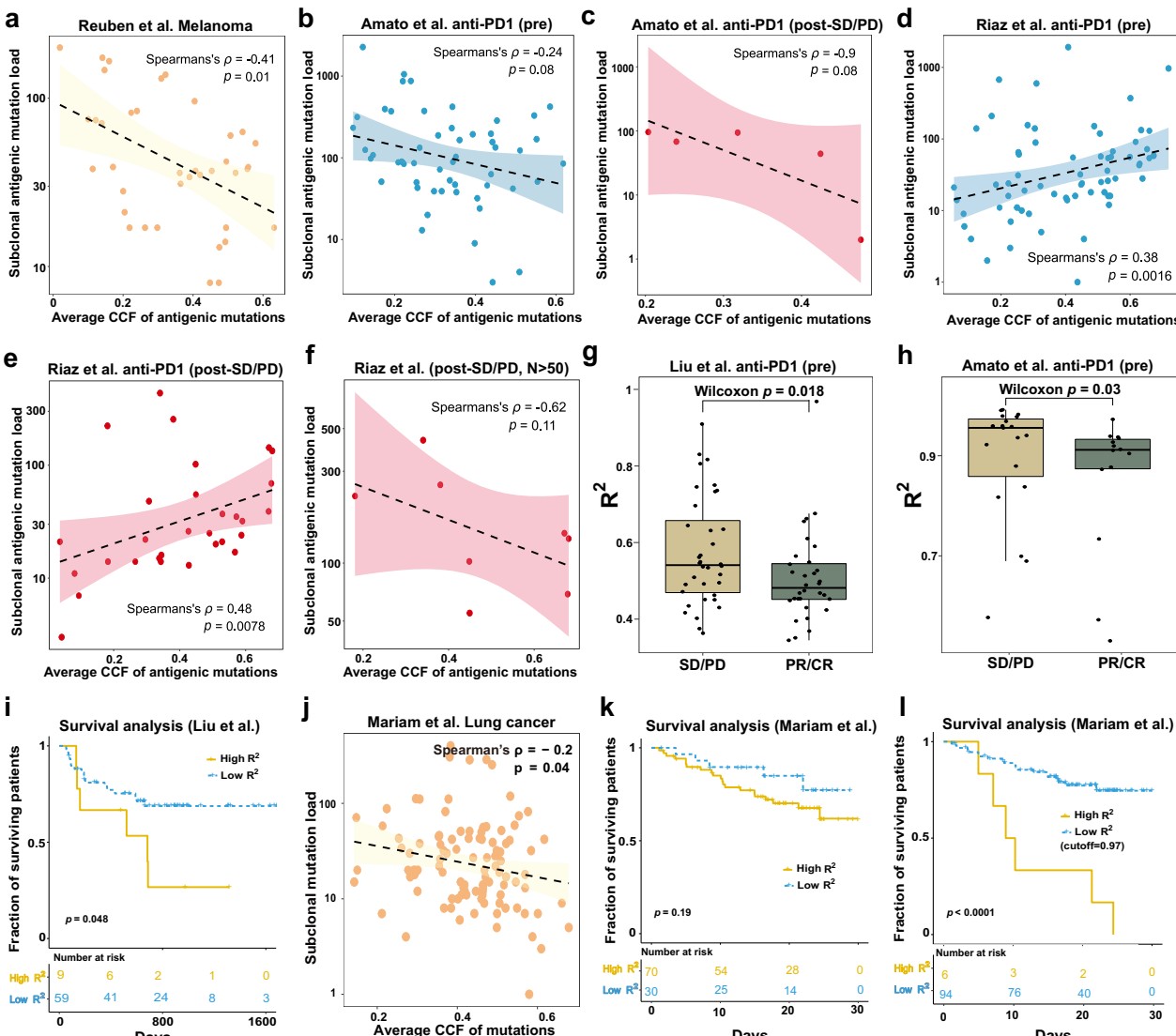

**Fig. 4 | Testing NFDS with whole-exome sequencing datasets.** Correlation analysis between the average CCF and number of subclonal antigenic mutations for samples ($n = 38$) from Reuben et al. (**a**), pre-therapy samples ($n = 54$) from Amato et al. (**b**), post-therapy SD/PD samples ($n = 5$) from Amato et al. (**c**), pre-therapy samples ($n = 67$) from Riaz et al. (**d**), post-therapy SD/PD samples ($n = 30$) from Riaz et al. (**e**), and samples (n = 8) with high subclonal neoantigen load (more than 50) from Riaz et al. (**f**). The line indicates the linear regression and the shading indicates the 95% CI of the regression. Power-law model of neutrality fitting for responders (PR/CR) vs non-responders (SD/PD) from Liu et al.'s cohort ($n = 36$ for left box and

$n = 32$ for right box) (**g**) and Amato et al.'s cohort ($n = 19$ for left box and n = 14 for right box) (**h**), respectively. Box plots show median, quartiles (boxes) and range (whiskers). $p$ values, Wilcoxon rank-sum one-sided test. **i** Survival analysis of patients ($n = 68$) from Liu et al.'s cohort stratified by neutrality fitting score $R^2$ (cutoff = 0.7: high, $R^2 \geq 0.7$; low, $R^2 < 0.7$). **j** Correlation analysis between the average CCF and number of subclonal mutations for samples ($n = 106$) from Mariam et al. Survival analysis of patients from Mariam et al.'s cohort ($n = 100$) stratified by neutrality fitting score $R^2$ with cutoff of 0.7 (**k**) and 0.97 (**l**), respectively. $p$ values, log-rank test.

cohort, we found a negative correlation between subclonal nonsynonymous mutation burden and mean subclonal CCF (Spearman's $\rho = -0.2$, Fig. 4j). Survival analysis shows that patients with a more neutral-like profile showed a higher survival rate compared to those with low $R^2$ values (more selection-like) (Fig. 4k, l). In conclusion, our results indicate that the distribution of CCF in pre-therapy tumor samples can serve as a predictive marker for the response to ICB therapy in melanoma patients. This discovery provides a new evolutionary biomarker for prioritizing patients for immunotherapy.

## Discussion

As a model of balancing selection, NFDS has been proven to be a robust mechanism for maintaining genetic diversity within a population[44–46]. Its role has been posited to drive coevolution between biological antagonists[53,54]. The diversity of major histocompatibility complexes (MHC) is a famous example. Tasked with recognizing foreign antigens, MHC molecules show

an extremely high allelic diversity within the human population[55]. A popular theory suggests that NFDS is a major contributor to this extensive diversity[56], as pathogens evolve to avoid the most prevalent MHC in the immune system[57].

Although neoantigens and immune systems are antagonists in tumor evolution[30], the potential role of NFDS in this context remains under-explored. Current theory posits that neoantigens undergo conventional negative selection[9]. Furthermore, the extensive mutational diversity in tumors is often attributed to neutral evolution[23–26]. Although several studies have argued for stringent selection in cancer, most of them assume that each genotype has a steady fitness effect, regardless of its relative frequency in the population[58,59]. Our current study proposes and tests the hypothesis of NFDS-driven tumor evolution, where cancer cells with higher neoantigen abundance are toward elimination and cancer cells with lower neoantigen abundance exhibit higher fitness. Through stochastic modeling, we

demonstrate that tumors subject to NFDS exhibit high intra-tumor heterogeneity and unique antigenic mutation landscape, similar to effective neutral evolution. In particular, a negative association between the average clonality of neoantigens and the number of subclonal antigenic mutations represents a distinct hallmark of NFDS. We then provide the quantitative evidence that supports NFDS in melanoma patients.

Our findings provide important insights into our understanding of immunotherapy and could potentially guide the development of more effective treatment strategies. We propose a potential explanation for why tumors with a high neoantigen load often prove resistant to ICB therapies. Our evidence suggests that NFDS contributes to the evolution of heterogeneous tumors that subsequently exhibit reduced responsiveness to ICB therapies. Additionally, the neutral tail of the neoantigen CCF distribution could serve as an indicator of NFDS and potentially predict which tumors are undergoing NFDS. Another potential application is to inform tumor treatment strategies, such as the adaptive therapy approach, in which patients are repeatedly given the minimum dose necessary to prevent the growth of treatment-resistant tumor cells and to control tumor size[60]. Further study of the NFDS will hopefully make it possible to tailor treatment to individual patients using their specific sequenced biopsy and adaptive therapy strategy combined with neoantigen-specific therapies[7].

Our study, while valuable as a proof of concept, acknowledges the limitations by the adoption of the predator-prey model where we regard immunogenic cells as a certain type of prey that immune cells are inclined to. We present exploratory simulations considering neoantigens on a CCF level to be a closer representation of NFDS and observed non-contradictory results (Supplementary Fig. 28). We also acknowledge the limitations of including only four melanoma patient cohorts[16,22,33,34] and one non-small-cell lung cancer (NSCLC) validation cohort[35]. This small sample size may not fully capture the heterogeneity seen across diverse tumor types. In addition, as immunotherapy has primarily been employed in melanoma[16,22,33,34] and non-small cell lung cancer[35]—both recognized as high mutation tumors—studying tumors with a relatively low number of mutations (sSNVs and/or indels) also holds promise for exploring the role of other genomic events, such as structural variations, in immunotherapy response. Expanding the study to include a larger set of pan-cancer patient cohorts for hypermutated tumor would allow for a more general examination of NFDS in immune selection and a more comprehensive analysis of the evolutionary dynamics of tumors. Meanwhile, the identification of the effectively neutral evolution calls for better quantification of Neoantigen-Mediated Immunoediting[61]. It is also important to note the potential challenges in distinguishing between NFDS and other immune escape mechanisms, particularly when a patient has not received ICB therapy. However, with the growing prevalence of ICB therapy, NFDS may emerge as another hurdle in tumor treatment, warranting further research and solutions.

In summary, our theoretical framework of tumor evolution offers critical insights into the evolutionary dynamics that occur between cancer cells and the immune system. We substantiate these findings by validating them with data derived from actual patients. This research not only enhances our understanding of tumor growth and immune interaction but also provides a valuable perspective that could guide future studies and the development of effective treatment strategies.

## Methods
### Stochastic branching process of neoantigen evolution
A stochastic model was developed to capture the evolutionary dynamics of neoantigens under NFDS. The model is based on a branching process that simulates the exponential growth of cancer cells[9,62,63]. In each step of the model, tumor cells belonging to a lineage can proliferate at a rate of $b$ and survive at a basal rate of $b_0$ in the absence of negative selection pressure. The two daughter cells resulting from each division accumulate mutations (regardless of sSNVs or indels) at an overall rate of $p_m$, where $m$ represents the number of generated mutations ($m \sim \text{Poisson}(p_m)$). A fraction of these mutations have antigenicity assigned at rate $p$, drawn from an exponential

distribution, which determines their potential to be recognized as neoantigens.

To initialize the model, the founder cell was set to be antigenically neutral. This assumption was made to avoid the rapid extinction of tumor lineages in cases where the founder cell carries a highly immunogenic neoantigen that elicits a strong immune response (Supplementary Fig. 29). In the model, a tumor cell $i$ can die with a death rate of $d_{C_i}$ determined by the accumulated antigenicity $A_{C_i}$ and the intensity of negative selection, denoted by $s$ (where $s$ is a non-positive number). The cumulative antigenicity $A_{C_i}$ is defined as the sum of antigenicities $\sum a_k^i$ carried by the neoantigens in the cell, where $a_k^i$ represents the $k$-th antigen of cell $i$. If a cell is considered immunogenic, its basal survival rate is reduced based on the cumulative antigenicity and the intensity of negative selection within the tumor microenvironment. As the number of neoantigens in a cell increases, the likelihood of becoming an immunogenic cell in the lineage also increases. This effect was modelled by setting a threshold $c_1$ for the cumulative antigenicity of tumor cells. Once a tumor cell accumulates enough neoantigens to exceed the threshold, it is considered immunogenic, i.e.

$$\sum a_k^i > c_1 \tag{1}$$

The Gillespie algorithm[62] was used to simulate populations of cells. Each time a cell divided, it acquired $m$ new unique mutations following the Poisson distribution. The newly derived mutations were assigned as neoantigens at rate $p$, or as passengers (evolutionary neutral) at rate $1 - p$. Each antigenic mutation was assigned an antigenicity value (denoted $a_k^i$ for the $k$-th antigen in a given cell $i$) sampled from an exponential distribution with mean equals to 0.2[9]. The death rate of each cell was determined by the complement of the survival rate reduced by the immunogenicity of the tumor cell. Then we have

$$d_{C_i} = 1 - \left(1 + sA_{C_i}\right)b_0 \tag{2}$$

Tumor growth and mutation accumulation were simulated using different parameter sets to capture the effects of tumor mutational characteristics and tumor microenvironment. Tumor growth was simulated up to a maximum population size of 100,000 cells. To account for the effect of NFDS, a weighting mechanism was introduced into the cell death rate. Highly antigenic mutations with high allele frequency (CCF) are more susceptible to immune selection since they can occupy a larger portion in $A_{C_i} = \sum a_k^i$ and are more likely to meet the condition $A_{C_i} > c_2$. This, in turn, increases the likelihood of cell death. In this context, NFDS is defined as a mechanism whereby the presence of highly immunogenic antigens in a larger proportion of cells leads to an increased impact on the cell death rates, i.e.

$$s = \begin{cases} (-2, 0), & \frac{T_{A_C}}{T} > c_2 \\ 0, & \frac{T_{A_C}}{T} \le c_2 \end{cases} \tag{3}$$

Here $T_{A_C}$ is the number of immunogenic tumor cells, $T$ is the total number of tumor cells, and $c_2$ is the threshold for the effect of NFDS. $-2 \le s \le 0$ was used in all simulations[9].

Immune escape was modelled by stochastically setting $s$ to 0 for immune escaped cells. Since our focus was on ICB therapies, cancer cells were allowed to acquire immune escape with a probability of $p_e$. Once a cell acquired immune escape, its survival rate was fixed to the basal survival rate $b_0$ regardless of its antigenicity. Furthermore, its daughter cells can inherit immune escape, giving them a survival advantage. Immune escape (e.g., PD-L1 overexpression) occurred as a consequence of mutation and is independent of antigen presenting machinery[9,40], thus the probability of immune escape ($p_e$) remains constant and does not influence antigenic mutation accumulation in simulations[9,17,40]. Following the study suggesting that driver

gene hotspots are highly conserved and have relatively poor neoantigen presentation[64], we assigned a rate of $p_{driver} = 10^{-6}$ for subclonal driver event and set cell antigenicity as $0.2 * A_{C_i}$ for cells with newly acquired driver mutations.

Parameters were chosen to represent different tumor-immune environments and literatures was reviewed to estimate parameter values in our model. The following parameters were used in all simulations: $b = 0.5$[65,66], $b_0 = 0.4$, $\mu = 5$, $p = 0.1$, $p_e = 10^{-4}$ (probability of immune escape, if applicable)[9,42]. For the analyses where cells and tumors were classified as immunogenic or non-immunogenic, the cell and tumor immunogenicity thresholds $c_1 = 0.5$ and $c_2 = 0.5$ were used[9], unless otherwise stated. A default value of $s = -0.8$ was chosen for most simulations, unless otherwise stated.

To calculate the overall growth rate of the simulated tumors, we performed a linear fit on the simulated tumor growth data (Supplementary Fig. 7a). The result showed a good fit between the fitted curve and the simulation curve. Consequently, the exponential growth of the simulated tumors can be characterized as $T \approx T_0 e^{rt}$, where $T_0$ and $T$ represent the initial and final cell population sizes respectively, and $r$ represents the tumor growth rate. Therefore, we can estimate the tumor growth rate based on the fitted exponential growth model:

$$r \approx \frac{\ln \frac{T}{T_0}}{t} \quad (4)$$

### Simulation of VAF/CCF distributions

Tumor evolution was simulated, and each mutation of every simulated tumor was assigned a unique index. To replicate actual sequencing data, mutations with a cancer cell fraction (CCF) greater than 0.04 were retained, and sequencing errors caused by sequencing depth and purity were simulated. The VAF (CCF/2) was considered as true values, and a statistical model that recreates the sequence noise seen in actual data generated the observed VAF. A binomial distribution was utilized in this model:

$$D_{obs} \sim Bin(n = N, p = VAF) \quad (5)$$

Here $N$ is the read depth of a given site and $D_{obs}$ is the number of observed reads of a mutation. Mutations with $VAF = 0$ were filtered.

### Whole-exome sequencing (WES) datasets from four melanoma cohorts

In this study, the publicly available WES data of normal-tumor pairs from the same patient were collected by accessing the Sequenced Read Archive (SRA) database (https://www.ncbi.nlm.nih.gov/sra). These cohorts are as follows:

Liu et al. anti-PD1 melanoma cohort[22]: This cohort consisted of 144 melanoma patients who received anti-PD1 therapy while a subset of 64 patients had progressed from prior ipilimumab treatment. Pre-treatment samples from all 144 patients were included in our analysis. We also used RNA-seq data and cytolytic scores from this cohort for our analysis.

Riaz et al. anti-PD1 melanoma cohort[16]: This cohort consisted of 68 melanoma patients who received anti-PD1 therapy. Among these patients, 35 previously had received anti-CTLA4 treatment (ipilimumab) and the other 33 had not received anti-CTLA4 treatment. We analyzed pre-therapy and on-therapy samples from this cohort, totaling 68 pre-therapy tumor samples and 41 on-therapy tumor samples. The cytolytic scores were also obtained from the original study[16].

Reuben et al. melanoma cohort with multiple therapeutics[34]: This cohort consisted of 14 patients with synchronous melanoma metastases. Among them, four had received targeted therapy, six received immunotherapy (three for PD-1 blockade, two for CTLA-4 blockade, and one for a combination of PD-1 blockade and CTLA-4 blockade), and four were treatment-naïve. A total of 40 tumor samples were analyzed, with two samples excluded due to the absence of paired normal tissue.

Amato et al. anti-PD1 melanoma cohort[33]: This cohort comprised 49 patients with melanoma who had undergone immunotherapy. A total of 64 tumor samples were analyzed, including 54 pre-therapy samples and 10 on-therapy samples, after excluding 2 tumor samples for lack of paired normal tissue.

These cohorts have the best overall response (BOR) to anti-PD1 and anti-CTLA4 combination ICB immunotherapy from the original studies. The BOR was determined based on Response Evaluation Criteria in Solid Tumors (RECIST) version 1.1 criteria, with complete responders (CR) and partial responders (PR) classified as responders, and stable disease (SD) and progressive disease (PD) as non-responders. Patients with a mixed response or non-evaluable response were excluded from the study.

To explore the role of NFDS in immune evasion, we used a published validation cohort comprising 100 non-small cell lung cancers[35]. We obtained the processed mutations, including VAF ($f$) and sCNA data, which includes local copy number ($N_T$). The CCF was determined using the provided purity ($\rho$):

$$CCF = \frac{f}{m\rho}(\rho N_T + 2(1 - \rho)) \quad (6)$$

where $m$ is the inferred multiplicity of the mutation.

### sSNV and indels calling and neoantigen prediction

For three cohorts[16,33,34], whose raw sequencing reads were available in SRA, the sSNV and indel mutations were called using a uniform pipeline. For Liu et al.'s cohort[22], the raw sequencing reads were not available and the processed mutations from its supplementary were used. The sequencing reads was first aligned to the human reference genome (hg38) using Burrows–Wheeler Aligner (bwa; 0.7.17-r1188)[67]. These aligned reads were then processed using GATK (v4.2.6.1; MarkDuplicates, BaseRecalibrator, ApplyBQSR, Mutect2 and FilterMutectCalls)[68]. MarkDuplicates marked duplicate reads, and BaseRecalibrator and ApplyBQSR recalibrated base quality scores together. Mutect2 was used to call somatic sSNVs and indels for each tumor/normal pair. The germline resource (af-only-gnomad.hg38.vcf.gz) and a panel-of-normals generated by CreateSomaticPanelOfNormals, which takes multiple normal sample call sets generated by Mutect2's tumor-only mode, were used. These sSNVs and indels were finally filtered using FilterMutectCalls and annotated using ANNOVAR (v.20200608)[69].

HLA alleles were called using POLYSOLVER 1.0.0 (https://anaconda.org/compbiocore/hla-polysolver). These HLA alleles, along with the called mutations, were used to predict neoantigen using NeoPredPipe[51], a pipeline designed for neoantigen prediction and evaluation. Both sSNV and indels resulting in amino acid changes were considered; novel peptides of eight, nine, and ten amino acids were considered. Mutations were annotated using ANNOVAR with default parameters. We considered a peptide to be a neoantigen if its predicted affinity ranked 2% (this cutoff has been used by Lakatos et al.[9] and is recommended in the NetMHCpan[70]), compared to a set of random natural peptides to the patient's HLA types.

### CCF estimation and analysis

The CNA and CCFs were estimated for each sSNV/indel, and then CCF analysis for these samples was performed. Allele-specific absolute copy number, ploidy, and tumor purity were estimated using TitanCNA[71], a hidden Markov model-based method. For each patient, the germline heterozygous SNP at dbSNP 146 loci in the normal biopsy were identified using SAMtools[72] and SnpEff v5.1[73]. Read counts for 1,000 base pair (bp) bins across the genome for all tumor samples were generated using HMMcopy1.36.0[74]. Then TitanCNA was used to determine the allelic ratios at the germline heterozygous SNP loci in the tumor biopsy and the depth ratios between the tumor and normal biopsies in the bins containing those SNP loci. When estimating allele-specific absolute copy number profiles, we only included SNP loci within WES-covered regions. Then 12 runs of TitanCNA with 12 combinations of subclone number, purity, and ploidy

were performed. In 9 out of the 12 combinations, different numbers of subclones ($n = 1$, 2 and 3), purity (0.2, 0.4, and 0.6), and ploidy (2) were set. In the remaining 3 combinations, we set a different number of subclones ($n = 1$, 2, 3) combined with the purity and ploidy estimated by Sequenza[75]. Then one run was selected for each tumor biopsy based on manual inspection of the fitted results, prioritizing the results with a single subclone unless results with multiple subclones better visibly fit to the data. Three samples were excluded because the results were difficult to interpret.

Then the CCFs and their variation (95% confidence interval) for each sSNV/indel were estimated using CHAT v1.1[76]. CHAT estimates the CCF for each sSNV by adjusting its VAF based on local allele-specific copy numbers at the sSNV locus. Then the CCFs for all sSNVs in the autosomes were calculated using the sSNV frequencies and copy number profiles estimated from the previous steps. Finally, we adjusted the CCFs using tumor purity estimated by TitanCNA. We distinguish subclonal mutations by selecting the upper bound of the 95% CI to be less than 1.

### Identification of effectively neutral evolution

The $R^2$ by fitting the VAF distributions of subclonal mutations to the expected power-law distribution in neutral model was calculated[25]. For each sample or simulated tumor, the inverse cancer cell frequency ($1/f$) and the cumulative frequency were calculated. We then fitted the cumulative frequency to a linear regression and obtained $R^2$, the ratio of the sum of squares of the regression to the sum of squares of the deviations. The closer $R^2$ is to 1, the more likely the tumor follows neutral evolution. We then compared the $R^2$ between patients with poor prognosis and patients with good prognosis in the patient cohorts.

### Statistics and reproducibility

All statistical analyses were conducted using R version 4.1.3 or Python version 3.9.6. The Wilcoxon rank-sum test was used to assess differences between the two groups, while the Kruskal-Wallis test was employed for comparisons among more than two groups. Spearman's $\rho$ coefficient was used to quantify correlations between variables. Survival analyses were performed using the Kaplan-Meier estimator to generate survival curves, and the nonparametric log-rank test was used to compare the survival curves. All simulated and WES datasets are available through public repositories to support reproducibility.

### Data availability

All patient data analyzed in this article are publicly available at BioProject (https://www.ncbi.nlm.nih.gov/bioproject) with accession numbers PRJNA639866, PRJNA316754, and PRJNA359359. Other data described in this article were obtained from the original studies. Numerical source data for graphs and charts are available at https://doi.org/10.5281/zenodo.11350667.

### Code availability

The codes of model simulation, data analysis, and visualization of this study are available at https://github.com/Shaoqing1117/Frequency-dependent2024 and https://doi.org/10.5281/zenodo.11350667[77].

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

## Acknowledgements

We thank members of Zhou and Hu laboratories for constructive discussions. This work was supported by the National Natural Science Foundation of China (11971405 to D.Z., 82241236 & 32270693 to Z.H.), Guangdong Basic and Applied Basic Research Foundation (2021B1515020042 to Z.H.) and Fundamental Research Funds for the Central Universities (20720230023 to D.Z.).

## Author contributions

Z.H. and D.Z. designed the study. S.C. constructed the models and performed the simulation studies. D.X. analyzed the patient sequencing data. S.C., D.X., Z.L., J.W., Z.H., and D.Z. interpreted the data. Z.H. and D.Z. supervised the study. S.C., D.X., Z.H. and D.Z. wrote the manuscript. All authors read and approved the final manuscript.

## Competing interests

The authors declare no competing interests.
