## [Peer Review File · Communications Biology]

Reviewers' comments:

Reviewer #1 (Remarks to the Author):

The research entitled “Frequency-dependent selection of neoantigens fosters tumor immune escape and predicts immunotherapy response” explores the role of negative frequency-dependent selection (NFDS) in the presence and absence of immune evasion. The authors demonstrate the correlation between NFDS in hypermutated tumors and the subsequent immune evasion, which is associated with a diminished response to immune checkpoint blockade (ICB) therapy. The authors perform computational modelling of cancer cell fraction (CCF) of neoantigens and draw evidence from four distinct melanoma cohorts to corroborate their findings. While the study presents potential ground breaking insights, it also raises several methodological and conceptual concerns that require further exploration and validation.

1. Neoantigen Identity Preservation in Simulations:

- It's ambiguous whether, during the simulation, the identity of an early-acquired neoantigen is consistently preserved and passed down to subsequent generations, leading to the clonal proliferation of that specific neoantigen throughout the tumor population. Does a unique antigen that clonally expands have the same effect that a whole population of cells carrying multiple different neoantigens?

2. Tumor Growth Rates in NFDS vs. PS Tumors:

- It's puzzling why tumors under NFDS (IE) grow faster than those under PS. Logically, NFDS should be on par with PS in the initial stages when neoantigen clonality is minimal. This raises the question: is the rate of neoantigen acquisition per cell division set exceptionally high? The authors should consider exploring varying rates of neoantigen acquisition, beyond just altering the 's' parameter. Without neoantigens, one would expect both models to exhibit similar behaviors. Also, what happens with clonal escape? What if the ability to evade the immune system is acquired very early. As demonstrated in previous studies, immune escape should allow the accumulation of neoantigens making tumors more susceptible to immunotherapies not less (<https://doi.org/10.1038/s41588-023-01313-1>).

3. Exploration of Simulation Parameters:

- The authors should venture into simulations with diverse parameters. It would be insightful to understand which specific combination of these parameters causes a mutation rate between 5 and 6 to lead to the complete eradication of virtual tumors. Growth curves of simulated tumors are built upon the assumption that each cell division results in unique mutations, which could

be classified as neoantigens or passengers. Do they assume equal mutations rates in each round of division? For instance, if a mutation is affecting a gene involved in DNA repair (i.e., BRCA1), there is a higher probability of accumulating even more mutations in the following cycles. Also, authors should consider driver events that can be immunogenic (TP53 events, see <https://doi.org/10.1038/s41586-022-04696-z>)

4. Clarification on PS Model:

- Based on my interpretation, the PS model essentially represents a non-frequency-dependent selection model (pure negative selection labelled as purifying selection). If this is accurate, it would be more apt to label it only NS (negative selection) versus FDNS (frequency dependent negative selection). These models have also been explored recently, and therefore it is important to add the citations (<https://doi.org/10.1158/0008-5472.CAN-21-3717>)

5. Influence of Subclonal Immune Escape:

- The pronounced influence of subclonal immune escape on the antigenic landscape under PS, contrasted with its diminished role under NFDS, is intriguing. Further, observing that a substantial fraction of tumors possess multiple escape routes indicates that the PS model might be more widespread than its NFDS counterpart.

6. Graph Labeling for CCF:

- When illustrating the relationship between average CCF and antigenic mutation load, it's crucial to specify that the "average CCF" pertains to the average CCF of antigenic mutations.

7. Accumulation of High CCF Antigens in NFDS:

- The reasons behind NFDS (IE) not facilitating the accumulation of high CCF antigens remain unclear. My perception is that the exclusive subclonal escape model might have inherent biases. As tumors evolve, escape mechanisms will eventually become entrenched, making PS (IE) and NFDS (IE) virtually indistinguishable. This observation is evident in Supplementary Figure 10, where the fraction of escaped cells ranges from 0 to 1 in PS, but only from 0 to 0.1 in NFDS. The lack of data for higher prevalence of escaped cells is a concern.

8. Exploration of other tumor types

- Broadening the research scope to encompass other high TMB tumor types, such as lung, bladder, and colorectal cancers, would provide a more holistic understanding of NFDS's role in immune evasion. Also, the majority of tumors harbour low number of point mutations, can the authors discuss the the role of other genomic events in light of immunotherapy response. Overall, also the authors should consider amplifying the number of simulations, ensuring a more robust and comprehensive data set.

Reviewer #3 (Remarks to the Author):

In their work Chen & Xie and colleagues carry out a mathematical exploration of negative frequency dependent selection as the mechanism shaping the antigenic mutation landscape of tumours. Their model represents a not groundbreaking but important step forward in theoretical modelling of cancer-immune co-evolution. In my opinion the most significant contributions of this work are: (i) the identified pattern of negative vs positive correlation between antigenic mutation load and average CCF of antigenic mutations; (ii) their analysis of ICB treated dataset in light of their theoretical findings. However, it is not entirely clear how robust the identified patterns are. While the authors already carry out a thorough exploration of their model, I believe the article would benefit from a more unbiased presentation of the findings and limitations.

1) My main concern is that what the authors implement here – based on my understanding – is a special version of frequency dependent selection: in the classical definition frequency dependent selection is defined on the allele/variant level, where the frequency of a particular allele/variant within the population affects its fitness. Here, however, frequency-dependence is defined by the proportion of “immunogenic cells” within the population, i.e. what matters is what percentage of cells carries “enough” antigens, but those antigens can be different. Therefore, their definition is almost as if “immunogenic” and “non-immunogenic” were the two alleles on which selection acts. I believe that classical frequency dependent selection (where the frequency of each neoantigen is considered) would be a closer representation of the cancer-immune interaction, e.g. the mice experiments they reference. Nonetheless, the model does offer some valuable insights as it is, and would be a useful starting point for future work into frequency dependent selection. My suggestion is to explore the limitations of this approach, at least in the text but ideally through some exploratory simulations. And carefully tone down or rephrase statements that are misleading about this dynamic, e.g. implying that the frequency of a given neoantigen is evaluated in the model (e.g. line 128-129).

2) My other major concern is about the parameter sensitivity of results, especially regarding the importance of IE and benefit of ICB in NFDS (vs PS) cancers. The hypermutator rate $\mu=5.5$ appears to be a special case which just allows for NFDS but not PS tumour growth at $s=-0.8$, while at lower and higher levels they show the same dynamics (Figure S6). It is therefore not surprising that NFDS shows limited advantage arising from immune escape opposed to PS that cannot grow without immune escape. Similarly, the benefit gained from ICB is related to the tumour growth benefit of IE, and therefore not surprising to show a clear distinction of PS and NFDS at $\mu=5.5$.

However, at $\mu=5$ or $\mu=6$ it seems that PS and NFDS are more similar (or at $\mu=5.5$ but lower/higher s) – either both achieve exponential growth or get eliminated. Therefore, it is not clear whether the substantial differences in IE/ICB hold for these parameter values. To provide a complete picture it would be required to explore these regimes as well, and evaluate which conclusions (e.g. differences between PS and NFDS, effect of IE on growth, benefit from ICB) are dependent on the combination of μ and s .

Overall, I find the choice of $\mu=5.5$ for Fig. 2a-b & 3a is misleading: it is one particular parameter that showcases drastically different behaviour between PS and NFDS, but this behaviour is not consistent for other parameter choices – and otherwise $\mu=5$ is used for simulations generating the other figures (if I understood correctly). This should be discussed carefully in the text, and I would even recommend keeping $\mu=5$ consistently in main text figures and showing this peculiar parameter combination in the supplementary.

3) The identified negative correlation pattern between antigenic mutation load and average CCF seems to be parameter dependent as well (Fig. S9). I appreciate the authors did an exploration of other parameter values in the supplementary figures, but their discussion in the main text should emphasise whether findings are parameter dependent (e.g. trend is negative but no significant correlation at certain parameter values). Especially as this is a key finding that is used to analyse real sequencing data. I would suggest to simulate multiple cohorts of 100 tumours each and evaluate the range of ρ and p values obtained for each scenario/parameter combination.

How does this negative correlation (and positive correlation in PS) change/correspond to different values of s ?

Also, I believe that $CCF > 0.04$ is still quite low and a higher value should be used to compare theoretical results to real sequencing. For example, the range of average CCF values in simulated (Fig. 1e-f) and real data (Fig. 4a-f) are fairly different, but will probably be closer if using e.g. $CCF > 0.1$ – therefore it would be good to have theoretical results with this the threshold to contrast real data to.

4) The findings about R^2 test distinguishing PS/NFDS and being predictive in real data are

intriguing. However, I'm concerned that the R^2 value depends on both negative and positive selection; but also on choices made in sequencing and data processing: sequencing depth and purity (the choice of minimal CCF should only those mutations are considered that can be detected confidently and not lost due to limited resolution), number of mutations (as it affects the resolution of the CCF curve). It is unclear whether the "less-neutral" cases are showing positive or negative selection, or simply being affected by more sequencing noise. I appreciate the authors explore depth & purity, but it should also be shown that in the real data analysed (Figs. 4g & h), these don't differ between SD/PD and PR/CR. Additionally, could the authors visualise the CCF curves (especially in real data) to support the findings?

5) I find it intriguing that the CCF distribution in Fig. 3e would clearly show a subclone at $CCF=0.5$ in multiple tumours, because the CCF and size of the subclonal "bump" should depend on the time the positively selected mutation (immune escape) is introduced (Williams et al., Nature Genetics, 2018). If immune escape is introduced at a given tumour size, it is plausible, but if it is initiated stochastically, the subclone size should be more random – including cases when it is not observable, since the subclone has already taken over the whole cancer by the end of simulation. Were these five tumours specifically selected to make the point on the figure? Can the authors show/ comment on the full range of CCF distributions that is produced by simulated tumours?

6) The findings about neutrality and NFDS (especially testing neutrality to establish NFDS) raise the question of sensitivity to falsely identified neoantigens: false neoantigens that are not actually under negative selection would automatically "dilute" any selective signal and result in more neutral looking CCFs, even without NFDS. Can the authors explore in their simulations how false neoantigens would affect or ability to distinguish NFDS and PS? And in their analysis of real data, can they test if analysing neoantigens with stricter filters (e.g. strong binders) lead to the same conclusions?

7) Neutrality (R^2) based testing is evaluated in two cohorts, but not in the other two – is there a technical limitation for this? Similarly, is R^2 an effective predictor of survival in all cohorts? If possible, all analysis should be shown (in supplementary) and discussed to avoid potentially misleading the readers.

Minor comments:

Can the authors comment on whether tumour-immune dynamics would hold for longer simulations (up to a higher total number of cells)?

Are only antigenic mutations or all mutations (including neutral ones) considered in CCF, etc. analysis? I believe only antigenic mutations are shown in all panels, if this is correct, please clarify it in the beginning of the results main text.

It is unclear how the Shannon diversity of tumours is computed. For example, are sequencing limitations taken into account?

It was unclear to me if indel-based neoantigens were also considered in the analysis of real tumours or only sSNV-based ones?

Sentence on lines 164-166, "However, the cause of the same phenomenon" is unclear, maybe consider rephrasing?

Line 68: recognized by HLA molecules  presented by HLA

Line 119 typo: stachastic  stochastic.

Line 125 typo: respond even only one  respond even if only one

Line 135: tumor growth initiated  tumor growth is initiated

Line 261: results also were also  results were also

Line 289: neoantigens was  neoantigens were

Line 354: evidencethat  evidence that

Line 504: was used  were used

Line 510: sSNVsand  sSNVs and

Line 553: expectated  expected

Supplementary Fig. 11 caption: dentification  Identification

Cytolitic  cytolytic

Reviewer #1 (Remarks to the Author):

The research entitled “Frequency-dependent selection of neoantigens fosters tumor immune escape and predicts immunotherapy response” explores the role of negative frequency-dependent selection (NFDS) in the presence and absence of immune evasion. The authors demonstrate the correlation between NFDS in hypermutated tumors and the subsequent immune evasion, which is associated with a diminished response to immune checkpoint blockade (ICB) therapy. The authors perform computational modelling of cancer cell fraction (CCF) of neoantigens and draw evidence from four distinct melanoma cohorts to corroborate their findings. While the study presents potential ground breaking insights, it also raises several methodological and conceptual concerns that require further exploration and validation.

We thank the referee for the constructive comments. We agree that our work needs further exploration and validation. Since our mathematical modelling involves complex evolutionary process of tumors with many model parameters, it is crucial to understand how these parameters affect the results. Also, due to the indeterministic nature of stochastic simulations, massive simulations are required to guarantee the robustness of our results.

Below, we will respond point by point to the comments from the referee.

Note: As suggested by the reviewer, in the revised manuscript we use “NS” (negative selection) instead of “PS” to represent the purifying selection in our study.

1. Neoantigen Identity Preservation in Simulations:

- It’s ambiguous whether, during the simulation, the identity of an early-acquired neoantigen is consistently preserved and passed down to subsequent generations, leading to the clonal proliferation of that specific neoantigen throughout the tumor population. Does a unique antigen that clonally expands have the same effect that a whole population of cells carrying multiple different neoantigens?

We thank the referee for raising this question. We found the mathematical symbol sets in the **Results** and **Methods** section could be misleading. We have now updated the symbols in these sections to describe the simulation processes more clearly for readers.

In our simulations, all newly acquired mutations are inherited by daughter cells during cell division. Early-acquired neoantigens are consistently preserved and passed down to subsequent generations. As described in the Methods section, if a tumor cell i elicits immune predation, it dies with a death rate of d_{C_i} determined by the accumulated antigenicity A_{C_i} and the intensity of negative selection, denoted by s . The cumulative antigenicity A_{C_i} is defined as the sum of antigenicities $\sum a_k^i$ carried by the neoantigens in cell i , where a_k^i represents the k -th antigen. Then we have:

$$d_{C_i} = 1 - (1 + sA_{C_i})b_0.$$

As each a_k^i is sampled stochastically from an exponential distribution, A_{C_i} is different among cells thus varied cell death rate of d_{C_i} . As d_{C_i} is determined by the term sA_{C_i} , if the whole population of cells carrying multiple different neoantigens, cell death rates caused by the same negative selection s can be different. If the population carry the same unique antigen that clonally expands, then each cell dies with the same rate under a determined s .

2. Tumor Growth Rates in NFDS vs. PS Tumors:

- It's puzzling why tumors under NFDS (IE) grow faster than those under PS. Logically, NFDS should be on par with PS in the initial stages when neoantigen clonality is minimal. This raises the question: is the rate of neoantigen acquisition per cell division set exceptionally high? The authors should consider exploring varying rates of neoantigen acquisition, beyond just altering the 's' parameter. Without neoantigens, one would expect both models to exhibit similar behaviors. Also, what happens with clonal escape? What if the ability to evade the immune system is acquired very early. As demonstrated in previous studies, immune escape should allow the accumulation of neoantigens making tumors more susceptible to immunotherapies not less (<https://doi.org/10.1038/s41588-023-01313-1>).

We thank the referee for raising this point. We agree that the rate of neoantigen acquisition per cell division is set to be high in our simulations. We set a high neoantigen acquisition rate to explore the difference between NS and NFDS in highly immunogenic tumors. In the revised manuscript, we consider varying rates of neoantigen acquisition (**Fig. R1**). With a relatively low neoantigen acquisition rate and low s , we found no significant difference in tumor growth rates between NS and NFDS (**Fig. R1a**). NFDS tumors grow faster than NS tumors under higher neoantigen acquisition rate and stronger negative selection (**Fig. R1b and c**). We found similar trend in NS (IE) and NFDS (IE) tumors (**Fig. R2**). With subclonal immune escape, tumor growth is affected by both neoantigen acquisition rate and negative selection at early tumor progression stages. We also simulated NS and NFDS with clonal immune escape (**Fig. R3**) and found no significant difference in tumor growth rates.

In all, the difference in tumor growth rates under NS and NFDS is dependent on neoantigen acquisition rate and negative selection intensity s . With low neoantigen acquisition rate, strong negative selection can be ineffective, making both models (NS and NFDS) to exhibit similar behaviors. Negative selection leads to pronounced difference in tumor growth rates between NS and NFDS with high neoantigen acquisition rates. With clonal immune escape, the two models are indistinguishable regarding tumor growth.

As the NS model is still an effective model to explore tumor evolutionary dynamics and our results showed that subclonal immune escape makes tumors more susceptible to immunotherapies (**Supplementary Figs. 12c and 18a**), which is in agreement with previous study¹ (<https://doi.org/10.1038/s41588-023-01313-1>). However, with the NFDS

model we can explore how subclonal immune escape work in tumor antigen landscapes with different microenvironments. Our goal is to provide possible explanation for ICB therapy failure and find supportive evidence with the NFDS model.

Fig. R1 (Supplementary Fig. 1). Tumor growth rates with different neoantigen acquisition rate. a-c, Box plots showing tumor growth rates in 50 simulated tumors under NS and NFDS with neoantigen acquisition rate equals to 0.05 (a), 0.075 (b) and 0.1 (c). * represents a p value less than 0.05 and *n.s* represents no significant difference, one-sided Wilcoxon rank-sum test.

Fig. R2 (Supplementary Fig. 2). Tumor growth rates with different neoantigen acquisition rate. a-c, Box plots showing tumor growth rates in 50 simulated tumors under NS (IE) and NFDS (IE) with neoantigen acquisition rate equals to 0.05 (a), 0.075 (b) and 0.1 (c). * represents a p value less than 0.05 and *n.s* represents no significant difference, one-sided Wilcoxon rank-sum test.

Fig. R3 (Supplementary Fig. 3). Tumor growth rates with clonal escape. Box plots showing tumor growth rates in 50 simulated tumors under NS (IE) and NFDS (IE) with $p_e = 1$ to simulated clonal escape. *n.s* represents no significant difference, one-sided Wilcoxon rank-sum test.

3. Exploration of Simulation Parameters:

- The authors should venture into simulations with diverse parameters. It would be insightful to understand which specific combination of these parameters causes a mutation rate between 5 and 6 to lead to the complete eradication of virtual tumors. Growth curves of simulated tumors are built upon the assumption that each cell division results in unique mutations, which could be classified as neoantigens or passengers. Do they assume equal mutations rates in each round of division? For instance, if a mutation is affecting a gene involved in DNA repair (i.e., BRCA1), there is a higher probability of accumulating even more mutations in the following cycles. Also, authors should consider driver events that can be immunogenic (TP53 events, see <https://doi.org/10.1038/s41586-022-04696-z>)

We thank the referee for the suggestion to explore diverse simulation parameters. We now simulated varying negative selection and mutation rates under NS and NFDS, respectively (**Figs. R4 and R5**). The complete eradication of virtual tumors depends on the higher mutation rate and stronger negative selection imposed on them (**Figs. R4 and R5**). Notably, NFDS tumors exhibited robust evolutionary rescue at mutation rate equals to 5.5 with varying negative selection intensity s (**Figs. R4b and R5b**).

We assume equal mutation rates in each round of cell division to model tumors with high mutation rates such as microsatellite instable (MSI) tumors². MSI tumors usually contain MSH6/MSH3 inactivation and loss of DNA mismatch repair proficiency that drives hypermutator phenotype³. We do not model the process of inducing genomic instability by randomly accumulated mutations that involved in DNA repair. Instead, we consider the initialized founder cell in the simulations that have already acquired genomic instability with elevated mutation rate.

We conducted exploratory simulations in line with a study suggesting that driver gene hotspots are highly conserved and exhibit relatively poor neoantigen presentation⁴. We set a rate of $p_{\text{driver}} = 10^{-6}$ for subclonal driver event and set cell antigenicity as $0.2 * \sum a_k$ for cells with newly acquired driver mutations. We found evolutionary rescue in NS tumors with subclonal driver mutations at mutation rate $\mu = 5.5$ (**Fig. R4d**). Under stronger negative selection and higher mutation rate, tumors are still eliminated under negative selection pressure induced by high mutation rate (**Figs. R4d, e and R5d**). NFDS tumors are more susceptible to driver events at high mutation rate and stronger negative selection (**Fig. R5d**). Driver events may mask the effect of neoantigen accumulation to a certain extent but tumor elimination is not reversible with driver events at higher mutation rate and stronger negative selection.

Fig. R4 (Supplementary Fig. 9). Tumor growth curves with diverse parameters. a-c, Growth curves of 20 simulated NS tumors with mutation rate set to $\mu = 5$ (a), $\mu = 5.5$ (b) and $\mu = 6$ (c), respectively. **d-e,** Growth curves of 20 simulated NS tumors with the rate of driver events set to 10^{-6} under mutation rates of $\mu = 5.5$ (d) and $\mu = 6$ (e), respectively.

Fig. R5 (Supplementary Fig. 10). Tumor growth curves with diverse parameters. a-c, Growth curves of 20 simulated NFDS tumors with mutation rate set to $\mu = 5$ (**a**), $\mu = 5.5$ (**b**) and $\mu = 6$ (**c**), respectively. **d,** Growth curves of 20 simulated NFDS tumors with the rate of driver events set to be 10^{-6} under mutation rate of $\mu = 6$.

4. Clarification on PS Model:

- Based on my interpretation, the PS model essentially represents a non-frequency-dependent selection model (pure negative selection labelled as purifying selection). If this is accurate, it would be more apt to label it only NS (negative selection) versus FDNS (frequency dependent negative selection). These models have also been explored recently, and therefore it is important to add the citations (<https://doi.org/10.1158/0008-5472.CAN-21-3717>)

We appreciate the great suggestion for a very suitable label for the PS model in our work. We now name it as NS for all PS scenarios in our paper. We have added citation of this article (<https://doi.org/10.1158/0008-5472.CAN-21-3717>) in the **Discussion** section.

5. Influence of Subclonal Immune Escape:

- The pronounced influence of subclonal immune escape on the antigenic landscape under PS, contrasted with its diminished role under NFDS, is intriguing. Further, observing that a substantial fraction of tumors possess multiple escape routes indicates that the PS model might be more widespread than its NFDS counterpart.

We agree that the NS model is more widespread than the NFDS model. While the NS model can support many current studies^{1,5,6}, existing data suggest that it is imperative to study mathematical models considering intratumoral neoantigen heterogeneity and its corresponding microenvironment^{5,7}. In our work, we introduce the NFDS model to supplement existing models and investigate the underlying mechanisms contributing to the failure of ICB therapy. The NFDS model offers a potential explanation for ICB therapy failure, particularly in cases where tumors exhibit a substantial mutation burden and high T cell infiltration but remain unresponsive to ICB therapy (**Supplementary Fig. 23a-b**). In our view, uncovering evidence in patient data that supports the existence of the NFDS model and its predictive capability for ICB prognosis provides crucial insights into tumor evolutionary dynamics and represents a significant stride toward optimizing immunotherapy protocols.

6. Graph Labeling for CCF:

- When illustrating the relationship between average CCF and antigenic mutation load, it's crucial to specify that the "average CCF" pertains to the average CCF of antigenic mutations.

We appreciate the referee's suggestion to clarify that the term "average CCF" refers specifically to the average CCF of antigenic mutations. Indeed, this graph labeling could be misleading. Consequently, we have now explicitly indicated whether the average CCF pertains to antigenic mutations or encompasses all mutations in our figures.

7. Accumulation of High CCF Antigens in NFDS:

- The reasons behind NFDS (IE) not facilitating the accumulation of high CCF antigens remain unclear. My perception is that the exclusive subclonal escape model might have inherent biases. As tumors evolve, escape mechanisms will eventually become entrenched, making PS (IE) and NFDS (IE) virtually indistinguishable. This observation is evident in Supplementary Figure 10, where the fraction of escaped cells ranges from 0 to 1 in PS, but only from 0 to 0.1 in NFDS. The lack of data for higher prevalence of escaped cells is a concern.

We thank the referee for raising this point regarding the prevalence of escaped cells. To address this problem, we investigated the fraction of escaped cells in *virtual* tumors across different population sizes. Initially, we evaluated NS and NFDS models under the same population size of 100,000 cells. Our simulations indicate that the fraction of escaped cells is associated with the tumor population size under NFDS (**Fig. R6**). Additionally, we observed a correlation between the average CCF of antigenic mutations and the fractions of escaped cells in virtual tumors (**Supplementary Fig. 17**), although the correlation is weaker for the NFDS model. To achieve a higher prevalence of escaped cells, larger NFDS tumors are required.

The fraction of escaped cells can reflect the expression level of checkpoint ligands, which is one of the biomarkers including tumor mutation burden (TMB) and microsatellite instability (MSI)¹. The difference between NS and NFDS in the relationships of high CCF neoantigens and prevalence of escaped cells indicates technical limitations of these biomarkers.

Fig. R6 (Supplementary Fig. 13). Prevalence of escaped cells. Box plots showing fractions of escaped cells in NFDS tumors with varying predefined population sizes.

8. Exploration of other tumor types

- Broadening the research scope to encompass other high TMB tumor types, such as lung, bladder, and colorectal cancers, would provide a more holistic understanding of NFDS's role in immune evasion. Also, the majority of tumors harbour low number of point mutations, can the authors discuss the role of other genomic events in light of immunotherapy response. Overall, also the authors should consider amplifying the number of simulations, ensuring a more robust and comprehensive data set.

We thank the referee for raising this point. Regarding the issue of tumor types, we have included a lung cancer dataset (PMID: 28445112)⁸ in the revised manuscript. We found a negative correlation between average CCF and subclonal nonsynonymous mutations,

which is in line with the NFDS model predictions (Spearman's $\rho = -0.2$, $p=0.04$, **Fig. R7a**). We used subclonal nonsynonymous mutations here as a proxy for antigenic mutations, because the raw reads of this study⁸ are not publicly available, so we cannot predict neoantigens for them. Survival analysis shows that neutral-like patients (larger R^2 values) tended to have worse survival than patients with selection-driven patients (low R^2 values) (**Fig. R7b**). We have also amplified the number of simulations to 100 times each to conduct all the correlation analyses in our study.

Fig. R7 (Fig. 4j-l). **a**, Correlation analysis between the average CCF and number of subclonal mutations for samples from Mariam et al. **b**, Survival analysis of patients from Mariam et al.'s cohort stratified by neutrality fitting score R^2 (cutoff = 0.7: high, $R^2 \geq 0.7$; low, $R^2 < 0.7$).

In addition, since in this study we focus on the neoantigens caused by somatic mutations (somatic SNV and indels), and these mutations are correlated with clinical response to cancer immunotherapy (PMID: 30550719), the role of other genomic events in terms of immunotherapy response is beyond the scope of this study. However, we appreciate the perspective of the role of other genomic events in light of immunotherapy response. Since immunotherapy has been mainly used in melanoma⁹⁻¹² and non-small cell lung cancer⁸, both of which are high mutation tumors. By studying tumors with a relatively low number of SNV mutations, it is also promising to explore the role of other genomic events, such as structural variation in immunotherapy response.

Reviewer #3 (Remarks to the Author):

In their work Chen & Xie and colleagues carry out a mathematical exploration of negative frequency dependent selection as the mechanism shaping the antigenic mutation landscape of tumours. Their model represents a not groundbreaking but important step forward in theoretical modelling of cancer-immune co-evolution. In my opinion the most significant contributions of this work are: (i) the identified pattern of negative vs positive correlation between antigenic mutation load and average CCF of antigenic mutations; (ii) their analysis of ICB treated dataset in light of their theoretical findings. However, it is not entirely clear how robust the identified patterns are. While the authors already carry out a thorough exploration of their model, I believe the article would benefit from a more unbiased presentation of the findings and limitations.

We thank the referee for the constructive comment. We acknowledge that the primary contributions of this work lie in the discovery of distinct correlations between the antigenic mutation load and the average CCF of antigenic mutations, linked to varying CCF distributions of all mutations that serve as predictors for responses to ICB therapy. The crucial application of our theoretical model is evident in the validation of these results across datasets. In the comparison between NS and NFDS, we intentionally selected parameter sets that effectively differentiate these two forms of negative selection. Recognizing the need for a comprehensive understanding of the robustness of the identified patterns, we agree that a more unbiased presentation of the findings and an exploration of the limitations of our theoretical model in capturing biological processes are necessary.

1) My main concern is that what the authors implement here – based on my understanding – is a special version of frequency dependent selection: in the classical definition frequency dependent selection is defined on the allele/variant level, where the frequency of a particular allele/variant within the population affects its fitness. Here, however, frequency-dependence is defined by the proportion of “immunogenic cells” within the population, i.e. what matters is what percentage of cells carries “enough” antigens, but those antigens can be different. Therefore, their definition is almost as if “immunogenic” and “non-immunogenic” were the two alleles on which selection acts. I believe that classical frequency dependent selection (where the frequency of each neoantigen is considered) would be a closer representation of the cancer-immune interaction, e.g. the mice experiments they reference.

Nonetheless, the model does offer some valuable insights as it is, and would be a useful starting point for future work into frequency dependent selection. My suggestion is to explore the limitations of this approach, at least in the text but ideally through some exploratory simulations. And carefully tone down or rephrase statements that are misleading about this dynamic, e.g. implying that the frequency of a given neoantigen is evaluated in the model (e.g. line 128-129).

We thank the referee for raising this point regarding the definition of frequency-dependent selection in our work. The frequency-dependent selection employed here is based on the adoption of the predator-prey model. In this context, we consider immunogenic cells as a specific type of prey, and immune cells as predators inclined to interact with them¹³. Following this, we propose that the negative selection on immunogenic cells depends on their frequency. We have rephrased this statements in the **Results** section “Here highly immunogenic neoantigens that present in high clonality (measured by cancer cell fraction or CCF) have higher probability to meet both cutoffs we set, and thus will be more susceptible to negative selection” and in the **Methods** section “Highly antigenic mutations with high allele frequency (CCF) are more susceptible to immune selection for they can take up larger portion in $A_{C_i} = \sum a_k^i$ and are easier to satisfy the condition $A_{C_i} > c_2$, resulting in a higher likelihood of cell death.”

While we have assessed the frequency of specific highly antigenic mutations to some extent, we acknowledge that a more classical representation of frequency-dependent selection, considering the frequency of each neoantigen, would offer a closer approximation of the cancer-immune interaction. The low complexity and imprecise modeling of the biological definition of NFDS introduce limitations, as evidenced by the set parameters c_1 and c_2 . While the actual range of these parameters remains unknown, we have observed certain patterns through their manipulation. In our **discussion**, we address how these two parameters impact our results, aiming to prevent the inaccurate control of biological mechanisms (**Supplementary Figs. 5, 6 and 15**).

Here we present an exploratory simulation considering neoantigens on a CCF/variant level. For each cell, we set a threshold c for immune recognition. Only neoantigens that have $CCF > c$ are affected by negative selection and the cell death rate is calculated as:

$$d_{C_i} = 1 - \left(1 + s \sum a_k^i \right) b_0, \quad CCF_{a_k} > c$$

$$c \sim \text{uniform}(1\%, 10\%)^{14}$$

We obtain similar results from the simulation data (**Fig. R8**).

Fig. R8 (Supplementary Fig. 27). Exploratory simulations of revised NFDS. **a**, Growth curves of 20 simulated tumors under NS and NFDS ($\mu = 5.5$). **b**, Correlation analysis between average CCF and subclonal antigenic mutation load of 100 simulated NFDS tumors. Only neoantigens with $CCF > 0.1$ are used. The line indicates the linear regression and the shading indicates the 95% CI. **c**, CCF distributions of all mutations (including neutral passengers) of five simulated tumors under immune escape mode of NFDS.

2) My other major concern is about the parameter sensitivity of results, especially regarding the importance of IE and benefit of ICB in NFDS (vs PS) cancers. The hypermutator rate $\mu=5.5$ appears to be a special case which just allows for NFDS but not PS tumour growth at $s=-0.8$, while at lower and higher levels they show the same dynamics (Figure S6). It is therefore not surprising that NFDS shows limited advantage arising from immune escape opposed to PS that cannot grow without immune escape. Similarly, the benefit gained from ICB is related to the tumour growth benefit of IE, and therefore not surprising to show a clear distinction of PS and NFDS at $\mu=5.5$.

However, at $\mu=5$ or $\mu=6$ it seems that PS and NFDS are more similar (or at $\mu=5.5$ but lower/higher s) – either both achieve exponential growth or get eliminated. Therefore, it is not clear whether the substantial differences in IE/ICB hold for these parameter values. To provide a complete picture it would be required to explore these regimes as well, and evaluate which conclusions (e.g. differences between PS and NFDS, effect of IE on growth, benefit from ICB) are dependent on the combination of μ and s .

Overall, I find the choice of $\mu=5.5$ for Fig. 2a-b & 3a is misleading: it is one particular parameter that showcases drastically different behaviour between PS and NFDS, but this behaviour is not consistent for other parameter choices – and otherwise $\mu=5$ is used for simulations generating the other figures (if I understood correctly). This should be discussed carefully in the text, and I would even recommend keeping $\mu=5$ consistently in main text figures and showing this peculiar parameter combination in the supplementary.

We thank the referee for the suggestion to explore the parameter sensitivity of our results. In the case of $\mu = 5$, although both models grow exponentially, they have different growth rates (**Figs. 1d and R1d**), and the stronger the s , the greater the difference in evolutionary dynamics. Under strong negative selection ($s = -0.8$), We showed substantial differences in average CCF, antigenic mutation load and benefit from ICB (survival analysis) of NS and NFDS tumors (**Figs. 2-3**). The difference in tumor growth between NS and NFDS is even greater at $\mu = 5.5$ (**Fig. R4b and R5b**). We included $\mu = 5.5$ as an extreme case to emphasize the difference between NS and NFDS. In the case of $\mu = 6$, there seems to be no difference between the two models (**Supplementary Fig. 8**), and we agree that it is imperative to explore parameter sensitivity to completely understand the difference between NS and NFDS.

At $\mu = 5$, the benefit of IE for NS tumors becomes stronger with the increase of negative selection but not for NFDS tumors (**Fig. R9a-b**). As higher mutation rate leads to complete eradication of virtual tumors as s become stronger (**Fig. R4-5**), we consider the growth rates of these tumors as zero. At $\mu = 5.5$, the benefit of IE is still more pronounced in NS than in NFDS (**Fig. R9c-d**). At $\mu = 6$, we do not observe substantial difference in benefit of IE between NS and NFDS. Furthermore, the benefit of IE becomes more significant as μ increases from 5 to 6 at the same s (**Fig. R9**). The benefit from ICB evaluated with survival analysis is related to the tumor growth benefit of IE (**Fig. R10**). Overall, the difference in IE/ICB between NS and NFDS tumors persists for $\mu = 5$ and 5.5, but vanishes at $\mu = 6$. In the revised manuscript, we have kept $\mu = 5$ in the main text figures and showed $\mu = 5.5$ in the supplementary.

Fig. R9 (Supplementary Fig. 11). Effect of IE on tumor growth rate. a-b, Box plots showing tumor growth rates in 50 simulated tumors under NS (**a**) and NFDS (**b**) at varying s with mutation rate of $\mu = 5$. **c-d,** Box plots showing tumor growth rates in 50 simulated tumors under NS (**c**) and NFDS (**d**) at varying s with mutation rate of $\mu = 5.5$. **e-f,** Box plots showing tumor growth rates in 50 simulated tumors under NS (**e**) and NFDS (**f**) at varying s with mutation rate of $\mu = 6$.

Fig. R10 (Supplementary Fig. 19). Survival analysis representing the benefit from ICB. a-b, Kaplan-Meier curves of 50 virtual patients that receive immunotherapy under NS (a) and NFDS (b) with mutation rate of $\mu = 5$. **c-d,** Kaplan-Meier curves of 50 virtual patients that receive immunotherapy under NS (c) and NFDS (d) with mutation rate of $\mu = 5.5$. **e-f,** Kaplan-Meier curves of 50 virtual patients that receive immunotherapy under NS (e) and NFDS (f) with mutation rate and of $\mu = 6$.

3) The identified negative correlation pattern between antigenic mutation load and average CCF seems to be parameter dependent as well (Fig. S9). I appreciate the authors did an exploration of other parameter values in the supplementary figures, but their discussion in the main text should emphasise whether findings are parameter dependent (e.g. trend is negative but no significant correlation at certain parameter values). Especially as this is a key finding that is used to analyse real sequencing data. I would suggest to simulate multiple cohorts of 100 tumours each and evaluate the range of rho and p values obtained for each scenario/parameter combination.

How does this negative correlation (and positive correlation in PS) change/respond to different values of s?

Also, I believe that $CCF > 0.04$ is still quite low and a higher value should be used to compare theoretical results to real sequencing. For example, the range of average CCF values in simulated (Fig. 1e-f) and real data (Fig. 4a-f) are fairly different, but will probably be closer if using e.g. $CCF > 0.1$ – therefore it would be good to have theoretical results with this the threshold to contrast real data to.

We appreciate the suggestion for more robust results regarding the negative correlation under NFDS and the excellent idea to set a higher value of $CCF > 0.1$. The identified negative correlation is indeed parameter-dependent. We simulated 100 tumors for each value of s under NS and NFDS, collecting antigenic mutations with $CCF > 0.1$. The result exhibited a certain robustness of our conclusion (**Fig. R11**). The positive correlation in NS and the negative correlation in NFDS depend on the value of s (**Fig. R11a and b**). When evaluating the ρ and p values at each s, both correlations become more significant as the negative selection intensity be stronger. Weak negative selection ($s = -0.1$) can result in insignificant correlations. Additionally, we found similar results in IE tumors, where stronger negative selection causes more pronounced effect of IE (**Fig. R11c and d**).

Fig. R11 (Supplementary Fig. 16). Correlation analysis between average CCF and subclonal antigenic mutation load. **a-b**, Correlation analysis between average CCF and subclonal antigenic mutation load of 100 simulated tumors undergoing NS (**a**) and NFDS (**b**) at varying s (CCF>0.1). **c-d**, Correlation analysis between average CCF and subclonal antigenic mutation load of 100 simulated tumors undergoing NS (IE) (**c**) and NFDS (IE) (**d**) at varying s (CCF>0.1). The line indicates the linear regression and the shading indicates the 95% CI.

4) The findings about R^2 test distinguishing PS/NFDS and being predictive in real data are intriguing. However, I'm concerned that the R^2 value depends on both negative and

positive selection; but also on choices made in sequencing and data processing: sequencing depth and purity (the choice of minimal CCF should only those mutations are considered that can be detected confidently and not lost due to limited resolution), number of mutations (as it affects the resolution of the CCF curve). It is unclear whether the “less-neutral” cases are showing positive or negative selection, or simply being affected by more sequencing noise. I appreciate the authors explore depth & purity, but it should also be shown that in the real data analysed (Figs. 4g & h), these don't differ between SD/PD and PR/CR. Additionally, could the authors visualise the CCF curves (especially in real data) to support the findings?

We appreciate the suggestion. In the revised manuscript, we have taken the influence of depth and purity into account when estimating the CCF. The CCFs and their variation (95% confidence interval) for each sSNV/indel were estimated using CHAT v1.1. CHAT¹⁵ estimates the CCF for each sSNV by adjusting its VAF based on local allele-specific copy numbers at the sSNV locus. Then the CCFs for all sSNVs in the autosomes were calculated using the sSNV frequencies and copy number profiles estimated from the previous steps. Finally, we adjusted the CCFs using tumor purity estimated by TitanCNA¹⁶.

As suggested by the referee, we have visualized the CCF curves of interest in Figs. 4g & h (subclonal mutations) and simulations below (**Figs. R12-14**):

Fig. R12 (Supplementary Fig. 24). Visualized CCF curves for samples from Amato et al.

Liu et al. anti-PD1 (pre)

Fig. R13 (Supplementary Fig. 25). Visualized CCF curves for samples from Liu et al.

Fig. R14 (Supplementary Fig. 20). Visualized CCF curves for simulations.

5) I find it intriguing that the CCF distribution in Fig. 3e would clearly show a subclone at CCF=0.5 in multiple tumours, because the CCF and size of the subclonal “bump” should depend on the time the positively selected mutation (immune escape) is introduced

(Williams et al., Nature Genetics, 2018). If immune escape is introduced at a given tumour size, it is plausible, but if it is initiated stochastically, the subclone size should be more random – including cases when it is not observable, since the subclone has already taken over the whole cancer by the end of simulation. Were these five tumour specifically selected to make the point on the figure? Can the authors show/ comment on the full range of CCF distributions that is produced by simulated tumours?

We appreciate the suggestion. Immune escape is initiated stochastically in our simulations. Due to the randomness of stochastic simulations, instances may arise where the subclonal “bump” is not precisely at CCF= 0.5, and in a few cases, it may not be observable. In **Fig. 3e**, we specifically highlight five simulated tumors as representative examples to illustrate this variability. The generality of this observation/result was confirmed by fitting our simulation data to power-law distributions. As depicted in **Fig. 3f** and **Fig. R14**, due to stochasticity, a few cases under NS showed R^2 values close to 1 (≥ 0.95), indicating scenarios where the immune escaped subclone has already taken over the entire cancer, and the CCF distributions closely resemble power-law distributions. To ensure a fair assessment, the statistically significant difference in CCF distributions between NS and NFDS is noted.

6) The findings about neutrality and NFDS (especially testing neutrality to establish NFDS) raise the question of sensitivity to falsely identified neoantigens: false neoantigens that are not actually under negative selection would automatically “dilute” any selective signal and result in more neutral looking CCFs, even without NFDS. Can the authors explore in their simulations how false neoantigens would affect or ability to distinguish NFDS and PS? And in their analysis of real data, can they test if analysing neoantigens with stricter filters (e.g. strong binders) lead to the same conclusions?

We appreciate the suggestion. To rule out the possibility of different filters leading to different patterns, we examined the patterns in **Fig. 4a-f** directly using all subclonal mutations without neoantigen prediction (**Fig. R15**), and we found that this mirrors the pattern in **Fig. 4a-f**, suggesting that the pattern of subclonal neoantigen mutations also reflects the overall pattern of subclonal mutations. In our simulations, subclonal antigenic mutations are sampled from all mutations generated from each cell division, thus the number of antigenic mutations represents a portion of all subclonal mutations in virtual tumors. The identified correlations are not affected by falsely identified neoantigens.

We agree that falsely identified neoantigens can result in more neutral looking CCFs. We use all subclonal mutations in determining the R^2 of each virtual and real tumor where falsely identified neoantigens are taken into account. The validation of negative correlation between subclonal mutation load and the average CCF in cohort-level can help to distinguish NFDS and NS.

Fig. R15 (Supplementary Fig. 22). Immune infiltration in patients. a-f, Correlation analysis between the average CCF and number of subclonal mutations for samples from Reuben et al. (**a**), pre-therapy samples from Amato et al. (**b**), post-therapy SD/PD samples from Amato et al. (**c**), pre-therapy samples from Riaz et al. (**d**), post-therapy SD/PD samples from Riaz et al. (**e**), and samples with high subclonal neoantigen load (more than 50) from Riaz et al. (**f**).

7) Neutrality (R^2) based testing is evaluated in two cohorts, but not in the other two – is there a technical limitation for this? Similarly, is R^2 an effective predictor of survival in all cohorts? If possible, all analysis should be shown (in supplementary) and discussed to avoid potentially misleading the readers.

We appreciate the suggestion. Since we identified a significant positive correlation between antigenic mutation load and average CCF of the Riaz et al⁸. melanoma cohort (**Fig. 4d and e**), we concluded that this cohort does not confer to the NFDS model. We did an R^2 fitting on the Riaz et al⁸. melanoma cohort in our revised manuscript and found no significant difference in the R^2 of responders and non-responders (**Fig. R16c**). As for the Reuben et al¹⁰. melanoma cohort, the information of immunotherapy response is not clear, therefore it's not available for the power-law fitting and survival analysis. Also, patients' survival data is not available in the study of Amato et al⁹. although they did survival analysis in their original study.

Fig. R16 (Supplementary Fig. 23). Immune infiltration in patients. **a-b**, Box plots showing the cytolytic score of responders (PR/CR) and non-responders (SD/PD) from Riaz et al. (**a**) and Liu et al. (**b**), respectively. Box plots show median, quartiles (boxes) and range (whiskers). **c**, Power-law model of neutrality fitting for responders (PR/CR) vs non-responders (SD/PD) from Riaz et al.'s cohort. P values, one-sided Wilcoxon rank-sum test.

Minor comments:

Can the authors comment on whether tumour-immune dynamics would hold for longer simulations (up to a higher total number of cells)?

We appreciate the suggestion. Our exponential fit indicates that simulated tumors with 100,000 cells exhibit a stable pattern of exponential growth. Equations (4) and (7) from our ODE models in Supplementary Note suggest that the negative and positive correlations are independent of tumor population size. Consequently, we think that the study of tumors with exponential growth can be extended to longer simulations, maintaining the dynamics of tumor-immune interactions.

Are only antigenic mutations or all mutations (including neutral ones) considered in CCF, etc. analysis? I believe only antigenic mutations are shown in all panels, if this is correct, please clarify it in the beginning of the results main text.

We appreciate the suggestion. We used the CCF of antigenic mutations in correlation analyses and all mutations in CCF distributions and power-law fitting. Details are noted in the legends of each panel and the Methods section.

It is unclear how the Shannon diversity of tumours is computed. For example, are sequencing limitations taken into account?

We appreciate the suggestion. The Shannon diversity of antigenic mutations is computed with the formula below:

$$\text{Shannon diversity} = - \sum_i (p_i \cdot \log_2 p_i)$$

where p_i represent the frequency of each antigenic mutation i .

We do not consider sequencing limitations in computing Shannon diversities, as we did not compare this result to real data. Instead, we view it as a theoretical guideline for subsequent analyses.

It was unclear to me if indel-based neoantigens were also considered in the analysis of real tumours or only sSNV-based ones?

Both indel-based and sSNV-based neoantigens were considered in the analysis of real tumors.

Sentence on lines 164-166, "However, the cause of the same phenomenon" is unclear, maybe consider rephrasing?

We have rephrased as "However, the cause of the same rollback pattern was different for NFDS where low-cellularity neoantigens of immunogenic cells were maintained by unsustainable negative selection."

Line 68: recognized by HLA molecules  presented by HLA
Done.

Line 119 typo: stachastic  stochastic.
Done.

Line 125 typo: respond even only one  respond even if only one
Done.

Line 135: tumor growth initiated  tumor growth is initiated
Done.

Line 261: results also were also  results were also
Done.

Line 289: neoantigens was  neoantigens were
Done.

Line 354: evidencethat  evidence that
Done.

Line 504: was used  were used
Done.

Line 510: sSNVsand  sSNVs and
Done.

Line 553: expectated  expected
Done.

Supplementary Fig. 11 caption: dentification  Identification
Done.

Cytolitic  cytolytic
Done.

References

1. Zapata, L. *et al.* Immune selection determines tumor antigenicity and influences response to checkpoint inhibitors. *Nat. Genet.* **55**, 451-460 (2023).
2. Lakatos, E. *et al.* Evolutionary dynamics of neoantigens in growing tumors. *Nat. Genet.* **52**, 1057-1066 (2020).
3. de Wind, N. *et al.* HNPCC-like cancer predisposition in mice through simultaneous loss of Msh3 and Msh6 mismatch-repair protein functions. *Nat. Genet.* **23**, 359-362 (1999).
4. Hoyos, D. *et al.* Fundamental immune–oncogenicity trade-offs define driver mutation fitness. *Nature.* **606**, 172-179(2022).
5. Riaz, N. *et al.* Tumor and Microenvironment Evolution during Immunotherapy with Nivolumab. *Cell* **171**, 934-949.e916 (2017).
6. Le, D. T. *et al.* Mismatch repair deficiency predicts response of solid tumors to PD-1 blockade. *Science* **357**, 409-413 (2017).
7. Mandal, R. *et al.* Genetic diversity of tumors with mismatch repair deficiency influences anti-PD-1 immunotherapy response. *Science* **364**, 485-491 (2019).
8. Jamal-Hanjani, M. *et al.* Tracking the evolution of non–small-cell lung cancer. *N. Engl. J. Med.* **376**: 2109-2121 (2017).
9. Riaz, N. *et al.* Tumor and Microenvironment Evolution during Immunotherapy with Nivolumab. *Cell* **171**, 934-949.e916 (2017).
10. Amato, C.M. *et al.* Pre-Treatment Mutational and Transcriptomic Landscape of Responding Metastatic Melanoma Patients to Anti-PD1 Immunotherapy. *Cancers* **12** (2020).
11. Reuben, A. *et al.* Genomic and immune heterogeneity are associated with differential responses to therapy in melanoma. *NPJ Genom Med* **2** (2017).
12. Liu, D. *et al.* Integrative molecular and clinical modeling of clinical outcomes to PD1 blockade in patients with metastatic melanoma. *Nat. Med.* **25**, 1916-1927 (2019).
13. Hamilton, P. T., Anholt, B. R. & Nelson, B. H. Tumour immunotherapy: lessons from predator-prey theory. *Nat. Rev. Immunol.* **22**, 765-775 (2022).
14. Gejman, R.S. *et al.* Rejection of immunogenic tumor clones is limited by clonal fraction. *eLife* **7**, e41090 (2018).
15. Li, B. & Li, J.Z. A general framework for analyzing tumor subclonality using SNP array and DNA sequencing data. *Genome Biology* **15**, 473 (2014).

16. Ha, G. et al. TITAN: inference of copy number architectures in clonal cell populations from tumor whole-genome sequence data. *Genome Research* **24**, 1881-1893 (2014).

Reviewers' comments:

Reviewer #1 (Remarks to the Author):

The authors have addressed all my Major comments.

Som minor comments

Discussion Section

Correct typo on line 357 to "effective."

Methodology section

Stochastic branching process of neoantigen evolution

Provide insight into the rationale behind the assumption that the probability of immune escape remained constant and independent of the accumulation of antigenic mutations.

Whole-exome sequencing (WES) datasets from four melanoma cohorts

Provide a more detailed explanation of the Reuben et al. group's therapeutic options, as only general terms are currently included.

Clarify which treatment groups received anti-CTLA4 treatment to allow for better interpretation of the subsequent discussions.

sSNV and indels calling and neoantigen prediction

Along the text, it is clearly indicated that just single nucleotide variants resulting in a change of the amino acid were considered.

Justify the selection of the 2% threshold for predicted neoantigen affinity compared to a set of random natural peptides for patients' HLA types. Provide additional information to support this threshold selection.

Reviewer #3 (Remarks to the Author):

In their recent revision, Chen & Xie and colleagues addressed my main concerns regarding the nature of NFDS and presenting an unbiased account of dynamics. However, realising that their analysis of R^2 values is obtained from all (not only antigenic) mutations, I have some questions regarding that:

I find it surprising that R^2 values separate NFDS and NS tumours so well, especially since (if I understand correctly) R^2 values are based on all mutations, not only neoantigens. Lakatos et al. (ref 9 of the manuscript) have shown that cumulative CCF curves of all mutations follow the neutral power-law expectation, and CCF curves of neoantigens (alone) diverge from it.

My understanding of the discrepancy is that when the authors report non-neutral/less-neutral R^2 values in simulated cancers, they are always considering simulations with immune escape. In that case, what they essentially test is positive selection on the immune escaped subclone, occasionally leading to a characteristic "bump" in the CCF distribution. Therefore, differences in presence/absence of this bump (also reflected in the R^2 value) indicate the presence of a distinct immune escaped subclone that is just the right size to show up in the distribution.

So my concern still stands that this non-neutral CCF distribution depends on when the subclone appears and how strongly it is selected. I'd expect that at too low and too high selection, there's no signal (quasi-neutral CCF distribution). Similarly, that the non-neutrality also depends on time

between immune escape occurring and final time point. Consequently, the chosen end-point of simulations would influence the presence/absence of non-neutrality – e.g. if the tumours grow longer, immune escaped subclone might become clonal leading to neutral-looking CCF distributions. This is mirrored in the authors' observation that immune escaped cells become more prevalent at higher population sizes (Fig. S13) – confirming that immune escape in NFDS is under weak positive selection that takes longer to overtake the population.

Overall, my concern is about whether the observed difference in CCF distributions and R^2 values is robust with respect to the final time/tumour size of the simulations. If my understanding is correct, it would depend on both s and final population size. escaped subclones are well detectable. Is the difference between R^2 values for NFDS and NS universal, or only applies in a certain parameter range? Nonetheless, I appreciate that the authors detect a difference in real clinical cohorts, but how this difference is interpreted depends on the answer to this question.

Questions/suggestions:

- 1) Is my assumption correct that R^2 values are tested on virtual tumours with subclonal immune escape? If so, it should be stated, using the labels otherwise stated in the manuscript, NS vs NS (IE). If not, can the authors explain why there would be a difference between their results and Lakatos et al.'s?
- 2) Can the authors explore the effect of time-since-immune-escape, by running simulations with either (i) a range of final tumour sizes or (ii) introducing immune escape at a fixed point during tumour growth?
- 3) I find the interpretation that NFDS is an immune escape strategy quite confusing – this is not something the cancer evolves, it is a property of the environment. Maybe interpreting NFDS as effectively lower selection against immunogenicity (and also lower selection for genetic immune escape) is clearer?
- 4) In Figs. S20, S24 & S25, I think it would be more informative to show the $1/CCF$ cumulative distribution curves instead, ideally with R^2 values, since that is used to generate Fig. 4.
- 5) Generally, some of the patient-derived CCF curves seem to be very different from virtual tumour curves. Most of the Liu et al. cohort seems to be dominated by a clonal CCF peak, with barely any subclonal "tail". However, the power-law theory applies only to the $1/f$ tail – can the authors comment why results from the virtual tumours are still relevant here? Are there appropriate numbers of subclonal mutations in these cases?

Other comments:

6) In the correlation analysis in Fig. S16: since the results seem to have a high spread, I think it would be beneficial to carry out this analysis multiple times (e.g. 50 times 50 tumours each at $s=-0.1$, NFDS, etc.) to obtain a distribution of ρ and p values for each scenario.

7) The sentence in the abstract "Altogether, our study provides quantitative evidence supporting the theory of NFDS in cancer, arguing against completely neutral evolution." should be revised since the study does not compare NFDS to neutral evolution, and in fact argues for NFDS based on neutral-like R^2 values. As it stands, the work rather provides another possible mechanism through which neutral-looking tumours can emerge despite being subject to selection – providing an explanation for the high prevalence of neutral-looking tumours.

Reviewer #1 (Remarks to the Author):

The authors have addressed all my Major comments.

Thank you for the constructive comments and positive feedback.

Some minor comments

Discussion Section

Correct typo on line 357 to “effective.”

Corrected.

Methodology section

Stochastic branching process of neoantigen evolution

Provide insight into the rationale behind the assumption that the probability of immune escape remained constant and independent of the accumulation of antigenic mutations.

Thanks for raising this problem. To model the effect of immunotherapy, we specifically consider immune checkpoint ligand overexpression (probably caused by mutation) as the immune escape mechanism in our simulations. Such mechanism is independent of antigen presenting machinery thus does not influence cell antigenicity^{1,2,3}. Thus, constant probability of immune escape was assumed. We have added these insights behind our assumption in the **Methods** section of our revised manuscript.

Whole-exome sequencing (WES) datasets from four melanoma cohorts

Provide a more detailed explanation of the Reuben et al. group's therapeutic options, as only general terms are currently included.

Thanks for this suggestion, we have revised this in the **methods** section:

“This Reuben et al. cohort consisted of 14 patients with synchronous melanoma metastases. Among them, four had received targeted therapy, six received immunotherapy (three for PD-1 blockade, two for CTLA-4 blockade, and one for a combination of PD-1 blockade and CTLA-4 blockade), and four were treatment-naïve. A total of 40 tumor samples were analyzed, with two samples excluded due to the absence of paired normal tissue.”

Clarify which treatment groups received anti-CTLA4 treatment to allow for better interpretation of the subsequent discussions.

Thanks for this suggestion, we have revised this in the **methods** section:

“This cohort consisted of 68 melanoma patients who received anti-PD1 therapy. Among these patients, 35 previously also had received anti-CTLA4 treatment (ipilimumab) and other 33 hadn’t received anti-CTLA4 treatment.”

sSNV and indels calling and neoantigen prediction

Along the text, it is clearly indicated that just single nucleotide variants resulting in a change of the amino acid were considered.

Yes, we have revised our manuscript to clarify that only sSNVs and indels resulting in changes of the amino acids were considered for neoantigen prediction.

Justify the selection of the 2% threshold for predicted neoantigen affinity compared to a set of random natural peptides for patients' HLA types. Provide additional information to support this threshold selection.

Thanks for this suggestion. We used 2% threshold because this is default cutoff in NeoPredPipe (PMID:31117948, <https://github.com/MathOnco/NeoPredPipe>). This cutoff was also used in Lakatos et al (PMID: 32929288). In fact, 2% cutoff was chosen based on the recommendation provided in the NetMHCpan paper (PMID: 28978689), which is used in the NeoPredPipe. To enhance clarity, we have included this information in our **Methods** section:

“We considered a peptide to be a neoantigen if its predicted affinity ranked 2% (this cutoff has been used by Lakatos et al (PMID: 32929288) and is recommended in the NetMHCpan paper (PMID: 28978689)), compared to a set of random natural peptides to the patient’s HLA types.”

Reviewer #3 (Remarks to the Author):

In their recent revision, Chen & Xie and colleagues addressed my main concerns regarding the nature of NFDS and presenting an unbiased account of dynamics. However, realising that their analysis of R^2 values is obtained from all (not only antigenic) mutations, I have some questions regarding that:

I find it surprising that R^2 values separate NFDS and NS tumours so well, especially since (if I understand correctly) R^2 values are based on all mutations, not only neoantigens. Lakatos et al. (ref 9 of the manuscript) have shown that cumulative CCF curves of all mutations follow the neutral power-law expectation, and CCF curves of neoantigens (alone) diverge from it.

My understanding of the discrepancy is that when the authors report non-neutral/less-neutral R^2 values in simulated cancers, they are always considering simulations with immune escape. In that case, what they essentially test is positive selection on the immune escaped subclone, occasionally leading to a characteristic “bump” in the CCF distribution. Therefore, differences in presence/absence of this bump (also reflected in the R^2 value) indicate the presence of a distinct immune escaped subclone that is just the right size to show up in the distribution.

So my concern still stands that this non-neutral CCF distribution depends on when the subclone appears and how strongly it is selected. I'd expect that at too low and too high selection, there's no signal (quasi-neutral CCF distribution). Similarly, that the non-neutrality also depends on time between immune escape occurring and final time point. Consequently, the chosen end-point of simulations would influence the presence/absence of non-neutrality – e.g. if the tumours grow longer, immune escaped subclone might become clonal leading to neutral-looking CCF distributions. This is mirrored in the authors' observation that immune escaped cells become more prevalent at higher population sizes (Fig. S13) – confirming that immune escape in NFDS is under weak positive selection that takes longer to overtake the population.

Overall, my concern is about whether the observed difference in CCF distributions and R^2 values is robust with respect to the final time/tumour size of the simulations. If my understanding is correct, it would depend on both s and final population size. escaped subclones are well detectable. Is the difference between R^2 values for NFDS and NS universal, or only applies in a certain parameter range? Nonetheless, I appreciate that the authors detect a difference in real clinical cohorts, but how this difference is interpreted depends on the answer to this question.

Thank you for raising this important issue. Indeed, we agree that R^2 values depend on how immune escape influences tumor evolution, particularly the strength of selection and the presence of a distinct immune-escaped subclone. Our findings, as illustrated in **Fig. 2**, **Supplementary Figs. 2, 11, 13, 18 and 22**, indicate that immune escape has different influences on *virtual* tumors depending on the selection parameter s as well as the final population size. Additionally, **Supplementary Fig. 3** suggests that we cannot distinguish NS from NFDS under clonal immune escape. Under the circumstances where the parameters fail to distinguish NS from NFDS tumors, there would be no signal (quasi-neutral CCF distribution).

Overall, the difference between R^2 values for NFDS and NS is observed within a certain parameter range we have examined. It is necessary to test the robustness of R^2 values to distinguish NS from NFDS. We have revised the main text to clarify this.

Questions/suggestions:

- 1) Is my assumption correct that R^2 values are tested on virtual tumours with subclonal immune escape? If so, it should be stated, using the labels otherwise stated in the manuscript, NS vs NS (IE). If not, can the authors explain why there would be a difference between their results and Lakatos et al.'s?

The assumption that R^2 values are tested on virtual tumors with subclonal immune escape is correct. We have now stated this in the revised manuscript and labeled in **Fig. 3f** that these virtual tumors were under subclonal immune escape (IE).

- 2) Can the authors explore the effect of time-since-immune-escape, by running simulations with either (i) a range of final tumour sizes or (ii) introducing immune escape at a fixed point during tumour growth?

Thanks for this suggestion. To explore the effect of time-since-immune escape, we introduced subclonal immune escape at four time points during tumor growth. We compared CCF distributions with both clonal and subclonal immune escape. During simulations with subclonal immune escape, we randomly chose a tumor cell to undergo immune escape when virtual tumors reached population sizes of 2.5%, 7.5%, 25% and 75% of the predefined final population size (**Fig. R1**). Simulations were terminated once virtual tumors reached the

predefined population size.

We found that R^2 values of NS and NFDS tumors are indistinguishable if immune escape was clonal or late subclonal, where the escaped subclones were not strongly selected (**Fig. R1a and e**). R^2 values separate NFDS and NS tumors well if immune escape was introduced at early subclonal stages (**Fig. R1b-d**). Parameter s can also influence R^2 values as weak negative selection doesn't distinguish NS from NFDS (**Fig. R1f**). These results revealed a certain degree of robustness of R^2 values to distinguish NS from NFDS.

Fig. R1 (Supplementary Fig. 22) Identification of neutrality for virtual tumors with IE introduced at fixed points. a, Identification of neutrality with clonal escape. b-e, Identification of neutrality with subclonal immune escape introduced when virtual tumors reached population sizes of 2.5%, 7.5%, 25% and 75% of the predefined final population size, respectively. f, Identification of neutrality with subclonal immune escape introduced at 7.5% pop size at varying s . The R^2 represents degree of fitting to power-law distribution. P values, one-sided Wilcoxon rank-sum test.

3) I find the interpretation that NFDS is an immune escape strategy quite confusing – this is not something the cancer evolves, it is a property of the environment. Maybe interpreting NFDS as effectively lower selection against immunogenicity (and also lower selection for genetic immune escape) is clearer?

Thanks for this great suggestion. We interpreted NFDS as an immune escape mechanism based on the antigenic mutation accumulation pattern observed in **Fig. 1e-f** and the evolutionary rescue of NFDS in **Supplementary Fig. 8-10**, which suggest that the dynamics of virtual NFDS tumors resembles those of NS (IE) tumors. Also, **Fig. 2c-d** and **Fig. 3a-b** suggest that NFDS leads to higher neoantigen heterogeneity and lower average CCF of virtual tumors, which is a mechanism employed by cancers to escape immune selection. We agree that it is clearer to interpret NFDS as effectively lower selection against immunogenicity thus shielding low-frequency neoantigens, as well as lower selection for genetic immune escape. We have revised this statement as interpreting NFDS as lower selection against immunogenicity.

4) In Figs. S20, S24 & S25, I think it would be more informative to show the 1/CCF cumulative distribution curves instead, ideally with R^2 values, since that is used to generate Fig. 4.

We have updated **Supplementary Fig. 21, 26 and 27** as:

Fig. R2 (Supplementary Fig. 21). Visualized cumulative distribution of CCF and $1/f$ model fitting for simulation data.

Liu et al. anti-PD1 (pre): SD/PD

Liu et al. anti-PD1 (pre): PR/CR

Fig. R3 (Supplementary Fig. 26). Visualized cumulative distribution of CCF and $1/f$ model fitting for samples from Liu et al.

Fig. R4 (Supplementary Fig. 27). Visualized cumulative distribution of CCF and 1/f model fitting for samples from Amato et al.

5) Generally, some of the patient-derived CCF curves seem to be very different from virtual tumour curves. Most of the Liu et al. cohort seems to be dominated by a clonal CCF peak, with barely any subclonal “tail”. However, the power-law theory applies only to the 1/f tail – can the authors comment why results from the virtual tumours are still relevant here? Are there appropriate numbers of subclonal mutations in these cases?

Thanks for these comments. Firstly, when we performed the neutral test for the Liu et al. cohort, we only used patients with a sufficient number (>20) of subclonal mutations. Secondly, we apologize for the confusion caused by our previous CCF curve, as it was plotted using only subclonal mutations instead of all mutations. In addition, it is more appropriate to also visualize the VAFs because we calculate the R^2 using the VAF. After plotting the subclonal VAFs of the Liu et al. cohort (Fig. R5-6), we can see that the patients with higher R^2 show a distinct neutral tail, such as patient 118 and patient 6, while patients (such as patient 49) with lower R^2 don't show an obvious neutral tail, but all of them still have subclonal mutations. Therefore, the results of the virtual tumors are still relevant to real patient data.

Fig. R5. Visualized VAF distributions of subclonal mutations for samples from the Liu et al.'s cohort (PR/CR).

Fig. R6. Visualized VAF distributions of subclonal mutations for samples from the Liu et al.'s cohort (SD/PD).

Other comments:

5) In the correlation analysis in Fig. S16: since the results seem to have a high spread, I think it would be beneficial to carry out this analysis multiple times (e.g. 50 times 50 tumours each at $s=-0.1$, NFDS, etc.) to obtain a distribution of ρ and p values for each scenario.

Thanks for this suggestion. We have carried out 20 times 50 tumors each and obtained a distribution of ρ and p for each scenario. We think 20 times are enough because we have observed a relatively stable pattern in the distributions of ρ and p at varying s . The results exhibited a certain robustness of our identified correlations (**Fig. R7**). The distributions of ρ and p depend on the value of s . When evaluating the ρ and p distributions at each s , the correlations become more significant as the negative selection intensity be stronger. Weak negative selection intensity can result in insignificant correlations (**Fig. R7b**).

Fig. R7 (Supplementary Fig. 17) ρ and p distributions of correlation analysis. a, Box plots showing ρ of correlation analysis between average CCF and antigenic mutation load under NS, NFDS, NS (IE) and NFDS (IE). **b,** Box plots showing p of correlation analysis between average CCF and antigenic mutation load under NS, NFDS, NS (IE) and NFDS (IE). Box plots show median, quartiles (boxes) and range (whiskers).

7) The sentence in the abstract “Altogether, our study provides quantitative evidence supporting the theory of NFDS in cancer, arguing against completely neutral evolution.” should be revised since the study does not compare NFDS to neutral evolution, and in fact argues for NFDS based on neutral-like R^2 values. As it stands, the work rather provides

another possible mechanism through which neutral-looking tumours can emerge despite being subject to selection – providing an explanation for the high prevalence of neutral-looking tumours.

Thanks for this suggestion. We agree that the statement in the abstract should be revised. Indeed, not all neutral-looking tumors undergo completely neutral evolution, especially those with high tumor mutational burden (TMB). By focusing on NFDS and investigating immunotherapy resistance in tumors with high TMB, our study provides an explanation for the high prevalence of neutral-looking tumors. We have revised our statements in the revised manuscript.

References

1. Beatty, G.L. & Gladney, W.L. Immune Escape Mechanisms as a Guide for Cancer Immunotherapy. *Clinical Cancer Research* **21**, 687-692 (2015).
2. Lakatos, E. et al. Evolutionary dynamics of neoantigens in growing tumors. *Nat. Genet.* **52**, 1057-1066 (2020).
3. Alsaab, H.O. et al. PD-1 and PD-L1 Checkpoint Signaling Inhibition for Cancer Immunotherapy: Mechanism, Combinations, and Clinical Outcome. *Front. Pharmacol.* **8**, 561 (2017).

REVIEWERS' COMMENTS:

Reviewer #3 (Remarks to the Author):

The authors answered my questions in this most recent revision.

One minor comment that can be addressed during the final editorial process: state the selection coefficient s used when generating supplementary figures (if there is a default value, specify that and state when deviating from this).

Reviewer #3 (Remarks to the Author):

The authors answered my questions in this most recent revision.

One minor comment that can be addressed during the final editorial process: state the selection coefficient s used when generating supplementary figures (if there is a default value, specify that and state when deviating from this).

Thank you for the positive feedback. We have a default value for parameter s and have specified this in the **Methods** section. In addition, we have stated in the figures when s deviates from the default value.

“A default value of $s = -0.8$ was chosen for most simulations, unless otherwise stated.”